

# Calving relation for tidewater glaciers based on detailed stress field analysis

Rémy Mercenier[1], Martin P. Lüthi[1], and Andreas Vieli[1]

[1]Department of Geography, University of Zurich, Zurich, Switzerland

*Correspondence to:* R. Mercenier (remy.mercenier@geo.uzh.ch)

**Abstract.** Ocean terminating glaciers in Arctic regions have undergone rapid dynamic changes in recent years, which have been related to a dramatic increase in calving rates. Iceberg calving is a dynamical process strongly influenced by the geometry at the terminus of tidewater glaciers. We investigate the effect of varying water level, calving front slope and basal sliding on the stress state and flow regime for an idealized grounded ocean-terminating glacier and scale these results with ice thickness and

velocity. Results show that water depth and calving front slope strongly affect the stress state while the effect from variations in basal sliding is much smaller. An increased relative water level or a reclining calving front slope strongly decrease the stresses and velocities in the vicinity of the terminus and hence have a stabilizing effect on the calving front. We find that surface stress magnitude and distribution are determined by solely the water depth relative to ice thickness for simple geometries. Based on this scaled relationship for the stress peak at the surface, and assuming a critical stress for damage initiation, we propose a

simple and new parametrization for calving rates for grounded tidewater glaciers that is in good agreement with observations.

## 1   Introduction

Many ocean terminating glaciers in the Arctic are currently undergoing rapid retreat, thinning and strong accelerations in flow. These dynamic mass losses contribute to about half of the Greenland ice sheet's contribution to sea level rise (van den Broeke et al., 2009), and are expected to further increase in the future (Nick et al., 2013). The mechanism of iceberg calving is thereby

at the heart of these rapid dynamic changes of ocean terminating glaciers. However, the understanding of the involved processes as well as the capability of predictive flow models to represent calving are limited (Vieli and Nick, 2011; Straneo et al., 2013).

Tidewater glacier evolution is the result of an interplay between mass flux from upstream and the rate and size of calving events (Post et al., 2011). Both processes are strongly influenced by the geometry of the glacier surface, the glacier bed and the bathymetry of the proglacial fjord (Nick et al., 2009) as well as external forcings such as submarine melt due to heat advection

by ocean currents (Motyka et al., 2013; Straneo and Heimbach, 2013; Straneo et al., 2013; Howat et al., 2010; Carr et al., 2013) or changes in ice-mélange (Joughin et al., 2008; Amundson et al., 2010).

Iceberg calving is a dynamical process of material failure which occurs when the local stress field in the vicinity of the calving front exceeds the fracture strength of ice, driving the formation and propagation of cracks and eventually leading to the detachment of a block of ice from the glacier front. The local geometry and water level at the terminus determine the stress field

and thereby the fracture processes and the geometry evolution. Further, buoyancy forces of submerged ice and erosion from



subaqueous melt are expected to enhance near-terminus stress intensity and hence calving rates, while a reclining terminus should reduce extensional stresses.

Several empirical and semi-empirical parametrizations of the calving rate for different terminus geometries have been proposed. A simple empirical relationship of linearly increasing calving rate with water depth, based on observations of tidewater glaciers in Alaska, has been established, used and extended for different regions (Brown et al., 1982; Benn et al., 2007b). This approach depends on the local water depth at the terminus only and is not process-based, and therefore independent of glacier geometry and dynamics (Vieli et al., 2001). In contrast, the flotation calving criterion, proposed by Van der Veen (1996) and modified by Vieli et al. (2001), determines the position of the terminus by calving away all ice that is close to flotation. In this approach the calving rate is an emergent quantity resulting from ice flow dynamics. Benn et al. (2007a, b) generalized the flotation criterion by setting the terminus position at the location where crevasses penetrate below the water level. The crevasse depth is computed using the Nye (1957) theory which relies on the equilibrium between longitudinal stretching and overburden stress of the ice. This dynamic approach for calving allowed for successful reproduction of calving front variations of ocean-terminating glaciers in Greenland and Antarctica (Nick et al., 2010; Otero et al., 2010; Nick et al., 2013; Cook et al., 2014; Otero et al., 2017). However, the crevasse depth estimation lacks validation with field observations and is based on a snapshot of the stress balance, neglecting the pre-existence of cracks and their effect on the stress state of the glacier (Krug et al., 2014).

The main driving force for calving in the class of models discussed above is the horizontal deviatoric stress $\sigma'_{xx}$ in vicinity of the laterally-confined calving front. Its magnitude can be estimated from the difference of vertically integrated hydrostatic pressure within the ice and of ocean water at the calving front (Cuffey and Paterson, 2010). The resulting extensional stress within the ice depends on the ice thickness $H$ and the water depth $H_w$ at the calving front

$$\sigma'_{xx} = \frac{\rho_i g H}{4} \left( 1 - \frac{\rho_w}{\rho_i} \omega^2 \right), \tag{1}$$

where $\rho_i, \rho_w$ and $\omega = H_w/H$ are the ice density, water density and relative water depth (Tab. 1). This equation illustrates the square dependence of the horizontal extensional stress on relative water level at the terminus. However, it should be noted that this vertically integrated stress is not representative for the stress state near the surface of the terminus, and the meaning of such a 'depth averaged' longitudinal stress for local fracture, for example for assessing surface crevasse formation, and the calving processes is not clear.

Using the above longitudinal stress boundary condition the maximum height for which a grounded glacier with a dry calving front can sustain a stable vertical front is approximately $110\,\mathrm{m}$ when crevasse depth is computed according to the Nye (1957) theory and $221\,\mathrm{m}$ when the ice is considered as undamaged and without crevasses (Bassis and Walker, 2012). These proposed maximum stable heights assume a hypothetical depth-averaged yield strength of $1\,\mathrm{MPa}$. However, observations of the appearance of surface crevasses on glaciers in relation to the strain rate field suggest a much lower cohesive strength of glacier ice between $0.09$ and $0.32\,\mathrm{MPa}$ (Vaughan, 1993) for cold Antarctic ice streams, or as low as $0.05\,\mathrm{MPa}$ for a temperate Alpine glacier (Lliboutry, 2002). Thus, the aforementioned maximum stable heights for a dry calving front are likely over-estimated, while the physical significance of a depth-averaged stress for a fracture process remains cumbersome. Furthermore, the pres-



ence of water along the calving front influences this maximum stable height, as an increase of water depth for a constant ice

thickness reduces the stresses and hence tends to increase the stability of the glacier front. Thus, a thicker glacier must terminate in deeper water in order for its calving front not to exceed a certain stress limit and to remain stable (Bassis and Walker, 2012).

Calving termini can also be over-steepened by melt undercutting. Ice flow model results (Hanson and Hooke, 2000; O'Leary and Christoffersen, 2013) suggest that an increase of water depth leads to a higher rate of over-steepening development at the

calving front. Thus, higher stress intensities should be expected which lead to an increasing calving activity with water depth. However, model results seem to indicate that melt undercutting does not significantly affect calving rates (Cook et al., 2014; Krug et al., 2015). Conversely, a calving front inclined towards the inland is expected to be more stable than a vertical cliff.

The detailed geometry of the calving front is constrained by the strong influence of frontal thickness and water depth on the stress state. Observations and theoretical considerations indicate a tendency of increasing relative water level with increasing

thickness (Bassis and Walker, 2012). This implies that thick glaciers approach flotation at their front but for shallow water depth the constraints on geometry seem less clear.

The relationship between water depth, stress state, front geometry and related calving type is well illustrated at the example of Eqip Sermia, a medium size ocean terminating outlet glacier on the West Greenland coast. Figure 1 shows that this glacier is characterized by two distinct calving front lobes with contrasting geometries: The grounded northern lobe exhibits a $200\,\mathrm{m}$

high inclined calving face with slope angles exceeding $45°$ while the southern lobe features a vertical ice cliff of $\sim 50\,\mathrm{m}$ freeboard with a water depth of $\sim 100\,\mathrm{m}$ (Lüthi et al., 2016). These substantially different geometries lead to distinct velocity and stress regimes in the proximity of the calving front which also determine the type of calving. The high, grounded, inclined northern cliff collapses at timescales of weeks, releasing large ice masses of up to $10^6\,\mathrm{m}^3$ and generating $50\,\mathrm{m}$ tsunami waves (Lüthi and Vieli, 2016). In contrast, the southern part of the front calves smaller volumes of ice at intervals of several hours.

Motivated partly by the case of contrasting calving front geometries at Eqip Sermia, the aim of this study is to better understand the detailed flow and stress regimes in vicinity of the calving front of tidewater glaciers, including those that are far from flotation. Using a numerical model that solves the full equations for ice flow, we investigate the sensitivity to variations in front thickness and slope, the water depth, and the strength of the coupling to the bed which results from sliding processes. We perform these model-experiments on idealized geometries of grounded glaciers and succeed to explicitly express the results as

function of relative water depth.

Based on these model results, we derive a novel parametrization of calving rate that is validated against observations from Arctic tidewater glaciers. This parametrization only requires the relative water level and is based on a fit to the modeled stress field at the surface and an isotropic damage evolution relation.





## 2 Methods

### 2.1 Ice flow model and rheology

We use the finite-element library libMesh (Kirk et al., 2006) to implement the Stokes equations for continuum momentum and mass conservation

$$\nabla \boldsymbol{\sigma} + \rho_i \boldsymbol{g} = 0\,, \tag{2}$$

$$\nabla \boldsymbol{u} = 0\,, \tag{3}$$

where $\boldsymbol{\sigma}$ is the Cauchy stress tensor, $\rho_i$ the ice density, $\boldsymbol{g}$ the gravitational force vector and $\boldsymbol{u}$ the velocity vector. As we assume ice to be incompressible and isotropic, the Cauchy stress tensor can be decomposed into an isotropic and a deviatoric part $\boldsymbol{\sigma}'$

$$\boldsymbol{\sigma} = \boldsymbol{\sigma}' + \sigma_m \boldsymbol{I}\,, \tag{4}$$

where $\sigma_m = \frac{1}{3}\mathrm{tr}(\boldsymbol{\sigma}) = \frac{1}{3}\sigma_{ii}$ is the isotropic mean stress and $\boldsymbol{I}$ the identity matrix. The ice rheology is described as viscous power-law fluid (Glen's flow law), linking the deviatoric stress tensor $\boldsymbol{\sigma}'$ to the strain rate tensor $\dot{\boldsymbol{\varepsilon}}$

$$\boldsymbol{\sigma}' = 2\eta\dot{\boldsymbol{\varepsilon}}\,. \tag{5}$$

The effective shear viscosity $\eta$ is defined as

$$\eta = \frac{1}{2}A^{-\frac{1}{n}}\left(\dot{\varepsilon}_e + \kappa_\varepsilon\right)^{\frac{1-n}{n}} \tag{6}$$

where $\dot{\varepsilon}_e = \left(\frac{1}{2}\dot{\varepsilon}_{ij}\dot{\varepsilon}_{ij}\right)^{\frac{1}{2}}$ is the effective strain rate, $A$ the fluidity parameter, $n = 3$ the power-law exponent and $\kappa_\varepsilon$ is a finite strain rate parameter included to avoid infinite viscosity at low stresses (Greve and Blatter, 2009).

The model domain was discretized with square, isoparametric Quad9 elements. The accuracy of the solution was improved with adaptive mesh refinement near the calving front. The Stokes equations with the nonlinear rheology were solved with the PETSc nonlinear solver SNES to a relative accuracy of $10^{-4}$ (Balay et al., 2008).

### 2.2 Model geometry and scaling

We used a 2-dimensional version of the model to conduct the geometrical tests, as illustrated in Figure 2. The geometry is defined in a cartesian coordinate system with horizontal axis $x$ and vertical axis $z$ with origin at the sea level at the calving front (where $x = 0$). The ice moves from right to left. The idealized glacier geometry used in all model experiments consists of a block of ice resting on a flat bed with a characteristic length $L = 2000\,\mathrm{m}$ and a characteristic ice thickness $H = 200\,\mathrm{m}$. The domain was discretized with 20 elements in the vertical and 200 elements in the horizontal which, after mesh refinement, led to a spatial resolution of $2.5\,\mathrm{m}$ in the terminus area.

All numerical results are scalable with reference values for ice thickness $H_{\mathrm{ref}}$ and overburden stress $\sigma_{\mathrm{ref}}$ and are therefore independent of the geometrical extent. This validity of the scaling was tested by running the model for different ice thicknesses





which recovered identical flow and stress results. The velocity scale $u_{\mathrm{ref}}$ was chosen as the vertical surface velocity caused by uniaxial confined compression in pure shear of an ice block under its own weight (Cuffey and Paterson, 2010)

$$H_{\mathrm{ref}} = H\,,$$
$$\sigma_{\mathrm{ref}} = \rho_i g H \sim \left[0.009\,\mathrm{MPa\,m^{-1}}\right] H\,, \tag{7}$$
$$u_{\mathrm{ref}} = \frac{A H \sigma_{\mathrm{ref}}^n}{8(n+1)} \sim \left[1.7 \cdot 10^{-6}\,\mathrm{m^{-3}\,a^{-1}}\right] H^4\,.$$

The coordinates and the water depth at the calving front $H_w$ are scaled by the ice thickness $H_{\mathrm{ref}}$

$$\widehat{x} = \frac{x}{H_{\mathrm{ref}}}\,, \qquad \widehat{z} = \frac{z}{H_{\mathrm{ref}}}\,, \qquad \omega = \frac{H_w}{H_{\mathrm{ref}}}\,. \tag{8}$$

All stress and velocity components are scaled according to

$$\widehat{\sigma} = \frac{\sigma}{\sigma_{\mathrm{ref}}}\,, \qquad \widehat{u} = \frac{u}{u_{\mathrm{ref}}}\,. \tag{9}$$

## 2.3 Boundary conditions

The upper surface of the glacier was described as a stress-free surface boundary. Basal motion was parametrized with a slipperiness coefficient $C$ which relates the basal velocity $u_b$ with basal shear stress $\tau_b$ (Gudmundsson and Raymond, 2008; Ryser et al., 2014)

$$u_b = C\tau_b\,. \tag{10}$$

This boundary condition was implemented as a two-element layer with constant viscosity $\eta_s = h_s/C$ which was added at the bottom of the model domain representing the glacier. At the bottom of this sediment layer, a Dirichlet boundary condition with zero velocity ($u = v = 0$) was imposed. A sediment layer thickness of $h_s = 10\,\mathrm{m}$ was chosen, although tests with varying $h_s$ showed no significant differences. This simple approach allowed to capture the physical processes that are relevant to this study. In the case of vanishing basal motion the two-element layer was left away, and Dirichlet boundary conditions ($u = v = 0$) were
imposed directly at the bottom of the model domain representing the glacier.

At the calving front a normal stress boundary condition was imposed below the water level, while the surface above water was kept stress-free. The stress boundary condition thus reads

$$\sigma_n = \min(\rho_w g z, 0) \tag{11}$$

where $\sigma_n$ is the normal stress applied on the calving front (negative, i.e. compressive since $z < 0$ below water) and $\rho_w$ is water density (Tab. 1).

At the upstream boundary of the glacier domain velocities were fixed to zero. Additional modelling experiments showed that different values for this upstream boundary condition do not affect the results of the analysis.





## 2.4 Sensitivity analysis strategy

The stress state and flow field near the calving front is analysed in three suites of numerical experiments that investigate the effect of variations in relative water level $\omega$, the slope of the calving front, and basal motion.

The water level sensitivity experiments were performed for relative water levels $\omega = 0, 0.25, 0.5, 0.75, 0.85$ and $\omega_f = \frac{\rho_i}{\rho_w}$, where the last value is the relative water level at flotation. The calving front for this experiment was vertical, and the bottom boundary without sliding (i.e. zero velocity Dirichlet boundary condition). All these experiments were undertaken with both the density of ocean water ($\rho_w = 1028\,\mathrm{kg\,m^{-3}}$) and freshwater ($\rho_w = 1000\,\mathrm{kg\,m^{-3}}$).

    The calving front slope sensitivity experiments were performed on a geometry with the upper part of the calving front

reclining at various angles. The lower 25% of the calving front height was set vertical, and the upper part inclined at angles from $90°$, $75°$, $60°$ and $45°$, until it reached the maximum surface height, see Figure 2 for illustration. This particular geometrical setup was chosen to represent a simplified geometry of Eqip Sermia, which has a $50\,\mathrm{m}$ high vertical cliff at the bottom with a $45°$ inclined slope up to the top at $200\,\mathrm{m}$. For this experiment, the relative water level was set to $\omega = 0$ and the sliding velocity was set to zero.

The bed slipperiness sensitivity experiments were performed on a block geometry with vertical calving front and relative water level $\omega = 0.5$. The basal slipperiness coefficient $C$ was varied from 0 to $1000\,\mathrm{m\,MPa^{-1}\,a^{-1}}$ with $333\,\mathrm{m\,MPa^{-1}\,a^{-1}}$ increments. A slipperiness of $1000\,\mathrm{m\,MPa^{-1}\,a^{-1}}$ corresponds to a sliding speed of $300\,\mathrm{m\,a^{-1}}$ for a typical tidewater outlet glacier in Greenland with a driving stress of $0.3\,\mathrm{MPa}$.

## 2.5 Stress invariant combinations

Any criterion for fracture propagation or damage evolution should be independent of the choice of coordinate system, and can therefore be expressed as a function of the invariants and eigenvalues of the stress tensor. Hayhurst (1972) proposed a linear combination of three stress invariants to describe the creep rupture of ductile and brittle materials under multi-axial states of stress. The invariants chosen were maximum principal stress $\sigma_1$, first stress invariant $I_1 = \sigma_m = \frac{1}{3}\sigma_{ii}$, and the von Mises stress $J_2 = \sigma_e = \left(\frac{3}{2}\sigma'_{ij}\sigma'_{ij}\right)^{\frac{1}{2}}$ to form the stress combination

$$\chi_H = \alpha\sigma_1 + \beta\sigma_e + \gamma\sigma_m, \tag{12}$$

where the weights $\alpha$, $\beta$ and $\gamma$ fulfill the conditions

$$0 \leq \alpha, \beta, \gamma \leq 1, \tag{13a}$$
$$\alpha + \beta + \gamma = 1. \tag{13b}$$

The Hayhurst stress $\chi_H$ has been used as a criterion for the initiation and evolution of damage in several glaciological studies (Pralong et al., 2003; Pralong and Funk, 2005; Duddu and Waisman, 2012, 2013; Duddu et al., 2013; Mobasher et al., 2016).

    To investigate the full spectrum of possible stress states that lead to the initiation of damage, we investigated linear combinations of five stress invariants: $\sigma_1, \sigma_e, \sigma_m$, and additionally the third invariant of the stress tensor $I_3 = \det(\boldsymbol{\sigma})$ and third invariant



of the deviatoric stress tensor $J_3 = \det(\boldsymbol{\sigma'})$. This extended linear combination reads

$$\chi = \alpha \sigma_1 + \beta \sigma_e + \gamma \sigma_m + \phi I_3 + \mu J_3 \tag{14}$$

with weights $\alpha, \beta, \gamma, \phi$ and $\mu$ that fulfill the conditions

$$0 \leq \alpha, \beta, \gamma, \phi, \mu \leq 1, \tag{15a}$$

$$\alpha + \beta + \gamma + \phi + \mu = 1. \tag{15b}$$

We performed a sensitivity analysis based on the five stress invariants of Equation (14) by systematically varying the weights
with 0.1 increments (Eq. 15)

## 3   Results

### 3.1   Sensitivity analyses

All sensitivity experiment results shown in Figures 3, 5 and 6 exhibit similar velocity and stress patterns. In general, the
modeled velocities and stresses increase towards the calving front, with a local stress maximum at the surface that is located
less than one ice thickness upstream of the calving front. This zone of high stress extends diagonally down towards the calving
front where it has a second local maximum closely above the water level. For experiments with a relatively low water level, the
absolute maxima in stress are found at the bottom of the calving face.

### 3.1.1   Water level height

The depth of the water at the calving front significantly impacts the stress regime and consequently the ice flow pattern and
magnitude near the terminus. The effect of different water depths on the stress field is displayed as Hayhurst stress in Figures
3A and 3C (with parameters chosen according to Pralong and Funk (2005), Tab. 1).
For a reduction in the relative water level from $\omega = \omega_f$ to $\omega = 0$ the maximum Hayhurst stress at the surface increases from
$0.08$ to $0.42\,\sigma_{\mathrm{ref}}$ and the location of the stress peak at the surface moves from $0.1$ to $0.5\ H$ upstream of the front, whereas the
Hayhurst stress at the vertical calving front increases from $0.15$ to $0.81\,\sigma_{\mathrm{ref}}$. Interestingly, the local maxima at the front are
always located near the water level. Further, the position of the global stress maximum for low water levels (below 0.25) is
found at the bottom of the calving front instead of the surface (Tab. 2).
Figures 3B, 3D and 4 illustrate how velocities close to the calving front increase by more than one order of magnitude when
the water level is decreased from near flotation ($\omega = 0.85$) to shallow water ($\omega = 0.25$). Note that for all water depths the
velocities are only affected up to approximately 2.5 ice thicknesses upstream from the front.

Extrusion flow, a velocity pattern for which maximum horizontal velocity occurs below the surface (Waddington, 2010), is
clearly visible in Figure 4 in vicinity of the calving front for the low water level cases. This pattern of extrusion flow near the
terminus was also observed by Hanson and Hooke (2000) and Leysinger–Vieli and Gudmundsson (2004).





In summary, increasing relative water depth leads to decreased flow velocities and lower stresses and moves the peak of the Hayhurst stress at the surface closer to the front.

### 3.1.2 Calving front slope

Results from the sensitivity experiment on calving front slope displayed in Figure 5 show large variations in stresses and flow speeds. Maximum Hayhurst stresses are found at the bottom of the calving front for all cases ranging from $0.57$ to $0.81\,\sigma_{\text{ref}}$ for slope angles between $45°$ and $90°$ (Tab. 3). A second, local maximum occurs at the surface behind the end of the slope, but the magnitude strongly decreases with decreasing slope. The maximum velocity for a $45°$ slope is $\sim 4$ times smaller than for a vertical calving front (Fig. 5D). Thus, as the calving front gets steeper, the stresses as well as the velocities increase. Again, the peak in Hayhurst stress at the surface moves further upstream as the calving front is becoming more gentle and a further local stress maximum occurs along the sloped surface. Moreover, the velocities along the surface do not peak towards at the front corner as in the vertical front case, but rather towards the bottom of the sloped surface, which is another sign of extrusion flow.

### 3.1.3 Bed slipperiness

The flow and stress regimes of the idealized glacier are less sensitive to an increase of bed slipperiness coefficient. Figure 6 shows that increased bed slipperiness leads to a slight increase in flow velocity, and the affected zone at the surface extends from 3 ice thicknesses in horizontal distance from the front to 5 ice thickness. Increasing the bed slipperiness coefficient produces very little effect near the front but causes a substantial increase of the stresses further upstream. The differences in the magnitudes of the Hayhurst stress maximum at the surface are, however, relatively small compared to the variations from other sensitivity experiments. The locations of the stress maxima remain the same for all bed slipperiness sensitivity experiments. Moreover, the spatial distributions of Hayhurst stress and velocity remain qualitatively very similar throughout the domain for the different bed slipperiness coefficients, and differences are mostly apparent at the surface.

### 3.1.4 Bed and surface slope

In the modeling presented so far we used a glacier geometry with horizontal surface and bed. Consequently the driving stress and hence velocities and stresses far upstream from the calving front are close to zero. In reality glaciers have a sloping surface. Therefore, we repeated some of the above experiments on a simple glacier geometry with a sloped bed and surface, a fixed cliff height and no sliding. Bed and surface slopes were chosen as -5° and 5°, respectively. Figure 7 illustrates the results: A reclining slope at the surface (i.e. surface height increasing towards the inland) with a flat bed leads to higher stresses and velocities upstream of the calving front as compared to the flat surface. However, the stress maximum and its location in the vicinity of the calving front remains almost identical (Fig. 7E,F). Similar results are obtained for a reverse bed slope with a flat surface (Fig. 7A,B).





To summarize, the stress and velocity fields in vicinity of the calving front are only slightly altered for sloping bed and surface. It is, however, noteworthy that the reclining surface slope induces higher stresses near the surface which could potentially induce crevassing, and thus advect pre-damaged ice to the calving front.

### 3.2 Stress invariant combinations

The Hayhurst stress, typically used as the driving force for damage evolution (Pralong et al., 2003; Pralong and Funk, 2005; Duddu and Waisman, 2012, 2013; Duddu et al., 2013; Mobasher et al., 2016), is not the only possible combination of objective stress measures. Here we attempt a systematic analysis of the possible stress invariant combinations (Eq. 14) and the corresponding locations of the stress maxima along the glacier surface. We illustrate this analysis at the example of the block geometry without any water pressure ($\omega = 0$) in Figure 8 where all possible linear combinations of five stress invariants along

the surface are displayed. While the stress combinations show a wide variety of curves, the maximum achievable stress states are dominated by the von Mises stress $J_2$ and the maximum principal stress $\sigma_1$, which both contribute to the Hayhurst stress. Hence, these two stress invariants are likely the driving factors for material failure in vicinity of the calving front. An important aspect illustrated in Figure 8 is the horizontal position of the stress maximum which is limited to $x_{max} \simeq 0.7 H_{\mathrm{ref}}$. This analysis thus suggests that a zone with maximum crevasse opening cannot be located in greater distance from the calving front

than $x_{max}$ for an idealized glacier without pre-damaged ice.

The magnitudes and positions of the maximum stress invariant combinations for different relative water levels $\omega$ are shown in Figure 9 (blue area corresponds to Fig. 8). The maximum stress for dry conditions ($\omega = 0$) is located $\sim 0.7 H_{\mathrm{ref}}$ from the calving front with a maximum von Mises stress of $\sim 0.45 \sigma_{\mathrm{ref}}$, whereas a water level close to flotation ($\omega = 0.85$) leads to a stress maximum of $\sim 0.15 \sigma_{\mathrm{ref}}$ at $\sim 0.25 H_{\mathrm{ref}}$ from the calving front. Figure 9 clearly illustrates that water pressure at the

calving front exerts a stabilizing effect on the calving front by both lowering the stresses and decreasing the distance from the calving front at which the stress maximum is located.

### 4 Stress parametrization and calving relation

The similarity of stress distribution curves along the glacier surface for varying relative water levels (Fig. 3C) allows for an explicit parametrization of the stresses. With some simple assumptions on a damage evolution law, a calving rate parametriza-

tion can be derived that is expressed as a function of total ice thickness and relative water level. For simplicity we assume that surface crevasses open under tension, and therefore consider only the maximum principal stress at the surface for the stress parametrization.

### 4.1 Stress parametrization

The distribution along the glacier surface of scaled maximum principal stress $\widehat{\sigma}_{1s}$ is shown for different relative water levels in

Figure 10 (this is approximately the tensile stress along the surface, whereas Figure 3C shows Hayhurst stress). The similarity in shape of these stress curves allows for an approximate representation by a function that depends on the relative water level





$\omega$ alone

$$\widehat{\sigma}_{1s}(\widehat{x}) = a(\omega)\widehat{\widehat{x}}\exp(-\widehat{\widehat{x}}),\tag{16}$$

where $\widehat{\widehat{x}}$ is a stretched and shifted version of the scaled (by ice thickness) horizontal coordinate $\widehat{x}$. This stretching function is somewhat cumbersome and is given in Appendix A. The extensional stress reaches the maximum at $\widehat{\widehat{x}} = 1$ (setting the derivative of Eq. (16) to 0) with magnitude $\widehat{\sigma}_{1,m} = a(\omega)\exp(-1)$, and can be approximated by

$$\widehat{\sigma}_{1,m}(\omega) = 0.4 - 0.45\left(\omega - 0.065\right)^2$$
$$\simeq 0.4\left(1 - \frac{\rho_w}{\rho_i}\left(\omega - 0.065\right)^2\right), \qquad \text{and therefore}\tag{17a}$$
$$a(\omega) = 1.087 - 1.223\left(\omega - 0.065\right)^2.\tag{17b}$$

The maximum extensional stress has a form very similar to the mean deviatoric stress in Equation (1) and is $\sim 60\%$ higher. The scaled horizontal position of the stress maximum can be approximated by

$$\widehat{x}_m = 0.67\left(1 - \omega^{2.8}\right).\tag{18}$$

## 4.2 Analytical calving relation

Using the parametrizations of magnitude and position of the maximum extensional stress at the surface (Eqs. 17a and 18) the calving rate can be estimated under simple assumptions on crevasse formation.

One major assumption is that a deep crevasse forms at the location of the maximum tensile surface stress where the ice is weakened until failure. Such crevassing seems realistic as both observations and model results show the formation of huge crevasses (Pralong and Funk, 2005). When failure of ice is complete, all ice in front of the crevasse breaks off and a new calving front forms at the location of the crevasse. Therefore, the average calving rate $\bar{u}_c$ can be calculated as the distance of the stress maximum divided by the time to failure $T_f$. In dimensional coordinates this is

$$\bar{u}_c = \frac{\widehat{x}_m H_{\text{ref}}}{T_f}.\tag{19}$$

Assuming further that crevasse formation can be described by isotropic damage formation with damage variable $D$, the stress in the damaged material is $\tilde{\sigma} = (1-D)^{-1}\sigma$ (Pralong et al., 2003; Pralong, 2006). The isotropic damage evolution relationship employed here is

$$\frac{dD}{dt} = B\frac{(\sigma_0 - \sigma_{th})^r}{(1-D)^{k+r}},\tag{20}$$

where $B$ is the rate factor for damage evolution, $r$ and $k$ are constants, $\sigma_0$ is the stress in the work zone and $\sigma_{th}$ a stress threshold for damage creation. Integrating this relation over time, the time-to-failure, i.e. the time required for damage to evolve from an



initial value $D_0$ to a critical value $D_c$, reads

$$T_f = \frac{(1-D_0)^{k+r+1} - (1-D_c)^{k+r+1}}{(k+r+1)B(\sigma_0 - \sigma_{th})^r} . \tag{21}$$

We further assume that the stress in the work zone is the maximum tensile stress $\sigma_0 = \sigma_{1,m}$. Inserting the parametrizations for maximum tensile stress and stress maximum position (Eqs. 17b and 18) in the above relation yields

$$\bar{u}_c = \frac{\widehat{x}_m H}{T_f} = \left[ \frac{0.67\,(k+1+r)\,B}{(1-D_0)^{k+r+1} - (1-D_c)^{k+r+1}} \right] \left(1 - \omega^{2.8}\right) (\sigma_{1,m} - \sigma_{th})^r\, H .$$

The term in square brackets is constant, and after renaming it the effective damage rate $\tilde{B}$, the expression reads

$$\bar{u}_c = \tilde{B} \left(1 - \omega^{2.8}\right) \left( \left(0.4 - 0.45\,(\omega - 0.065)^2\right) \rho_i g H - \sigma_{th} \right)^r H \tag{22}$$

with parameter values $\tilde{B} = 37\,\mathrm{MPa}^{-r}\,\mathrm{a}^{-1}$ and $\sigma_{th} = 0.17\,\mathrm{MPa}$ which were determined from a comparison with data discussed below. The parameter value $r = 0.43$ is chosen according to Pralong and Funk (2005).

## 5   Discussion

### 5.1   Sensitivity analyses

The stress intensity and therefore ice deformation rates are decreasing as the relative water level increases due to the pressure exerted by the water at the calving front. This feature is already captured by the mean extensional stress in the force balance (Eq. 1), and in more detail in the parametrized maximum extensional stress (Eq. 17a), illustrated in Figure 10. In both cases the square dependence of the horizontal stress on relative water level controls fracture or damaging processes, the magnitude and rate of which depend linearly on the stress intensity.

In addition, the detailed modeling shows that the stress peak at the glacier surface moves upstream for lowering water level (Figs. 3 and 10), implying that crevasses are likely to open in greater distance from the calving front and leading to detachment of larger masses during calving.

A higher relative water level results in a more stable calving front (as emphasized by Bassis and Walker, 2012) which seems to be in contrast with the often-used relations which predict that calving rates increase with water depth (Brown et al., 1982; Meier and Post, 1987; Hanson and Hooke, 2000). In nature, however, glaciers terminating in deeper waters are also thicker and calve at higher rates as they experience higher absolute (unscaled) stresses. Furthermore, submarine frontal melting is likely to lead to higher calving rates by over-steepening of the front (O'Leary and Christoffersen, 2013), although the melt undercutting effect on calving rates seems to be limited (Cook et al., 2014; Krug et al., 2015).

Using fresh water instead of sea water at the calving front yields slightly higher stresses and velocities (Fig.3C,D). This difference can be explained by the reduced back pressure applied by fresh water on the calving front, which results from a lower water density.

The model results demonstrate that reclining calving fronts lead to lower velocities and stresses and thereby implicitly confirm that inclined calving fronts should reach larger stable heights than vertical cliffs, as observed for example at Eqip




Sermia (200 m high at 45°). This sensitivity analysis on front slope may, together with observational data on non-vertical calving fronts, provide constraints on parameters of ice resistance to failure. Further, the presence of extrusion flow along the reclining calving face of an idealized glacier was demonstrated. Such a velocity pattern has been observed and measured on an inclined slope at the northern front of Eqip Sermia (Lüthi et al., 2016) but is rarely discussed in modeling studies (Hanson and Hooke, 2000; Leysinger–Vieli and Gudmundsson, 2004).

Basal sliding leads to increased stresses at the surface throughout the computational domain. Thus, basal sliding may cause an onset of ice damaging and crevasse opening in a greater distance from the calving front (Fig. 6C). The velocity patterns in Figure 6B show that the influence of bed slipperiness is only apparent in the proximity of the calving front, even for high sliding coefficients. Moreover, stress distributions are almost identical for all bed slipperiness experiments, which implies that basal sliding has a negligible effect on the stability of the calving front. Basal sliding adds a constant velocity at the bottom of the domain rather than affecting the velocity gradients. This result does not include any spatial variation in bed slipperiness, which would likely be caused by including a water pressure dependent sliding relation. Effective pressure (the difference between ice normal stress at the bottom and water pressure) typically decreases towards the calving front for real glaciers with sloping surface, and may cause additional sliding towards the front, an effect that is not considered in this modeling effort.

For a sloping glacier surface the location and magnitude of the stress maximum in vicinity of the calving front remain almost identical, as shown in Figure 7. Similar results are obtained for a reverse bed slope with a flat surface, with a smaller influence on stresses and velocities than for the reclining surface. However, the effect on stresses and velocities upstream of the calving front is not visible for the reverse bed slope with a flat surface. This indicates that, for a glacier with a reclining surface slope, ice can potentially start damaging and forming crevasses at the surface far upstream from the calving front.

## 5.2 Calving relation

The proposed calving rate parametrization (Eq. 22) is simple and only requires two geometrical quantities: frontal ice thickness $H$ and water depth $H_w$. It exhibits many similarities with established calving parametrizations, but is formulated in terms of two quantities that are calculated by any ice flow model. It therefore is a drop-in replacement for other calving relations used in glacier models of different complexity.

Being based on modeled stress fields and a damage rheology, the proposed parametrization is physics-based, as opposed to the purely or semi-empirical nature of other approaches (e.g. Brown et al., 1982; Van der Veen, 1996; Benn et al., 2007b). The assumptions about the failure process are expressed through three parameters $\tilde{B}$, $\sigma_{th}$ and $r$ which can be determined by comparison with data.

The calving rate parametrization (Eq. 22) has some interesting properties which are illustrated in Figures 11 and 12. Holding constant the relative water depth, the absolute water depth or the ice thickness results in different calving laws:

- For constant relative water level $\omega$ the calving rate grows roughly like $\bar{u}_c \propto H^{1+r}$ (black and gray lines in Figure 11);

- For constant absolute water depth $H_w = \omega H$ a fit shows that roughly $\bar{u}_c \propto H^{1.25}$ (red and orange lines in Figure 11);





- For constant ice thickness the calving rate decreases with increasing relative water level (Fig. 12) roughly like

$$\bar{u}_c \propto \left(1 - \omega^{2.8}\right)\left(1 - 1.3\,\omega^2\right)^r \simeq \left(1 - \omega^{2.8}\right)\left(1 - \left(\frac{\rho_w}{\rho_i}\omega\right)^2\right)^r .$$

The predicted calving rate for a given water depth depends on the thickness of the glacier which is the result of the mass fluxes in the terminus area. In that sense, calving rates depend on the surface evolution and hence the upstream dynamics of the glacier.

## 5.3 Comparison with data

The calving rate parametrization (Eq. 22) contains three empirical parameters $\tilde{B} = 37\,\mathrm{MPa}^{-r}\mathrm{a}^{-1}$, $\sigma_{\mathrm{th}} = 0.17\,\mathrm{MPa}$ that were obtained by comparison with data, and $r = 0.43$ which is take from Pralong and Funk (2005). These parameter values are within the range of previous studies (Duddu and Waisman, 2012; Lliboutry, 2002; Vaughan, 1993).

To obtain these parameter choices, data on calving rate, ice thickness and water depth for a wide variety of tidewater glaciers in the Arctic were collected. Unfortunately, many studies report only width-averaged data on calving front geometry and calving rate which are not suitable for our proposed relation which relies on local stresses on a flowline. Only a limited set of point data on calving front geometries are available from the published literature from which total ice cliff thickness, water depth and calving rate can be obtained. In this study we used the values shown in Table 5 from diverse data sources for comparison.

Contours of calving rates calculated with Equation (22) are shown in Figure 13. Velocity data from Table 5 are shown for comparison, most of which stems from glacier fronts that are close to flotation. The maximum theoretical calving front height predicted by Bassis and Walker (2012) is indicated.

Figure 14 plots the same calving rate data against predictions from the parametrization. While a sizable spread of the data is visible, especially for low velocities, the general agreement shows that the parametrization is suitable to predict calving rates for many tidewater glaciers in the Arctic.

## 6 Conclusions

This study improves our knowledge on the influence of geometry and water depth on the stress and flow regimes in the vicinity of the calving front, and proposes a novel calving rate parametrization.

The magnitude of the stresses and flow speeds near a grounded vertical calving front are dominantly dependent on water depth, and increase with decreasing water depth. Thus, the presence of water at the calving front has a strong stabilizing effect. Importantly, the extensional stress at the surface can be parametrized as a function of relative water level only. Further, we find that grounded tidewater glaciers with reclining calving faces have the potential to reach larger maximum stable heights than those with vertical calving fronts. Basal sliding likely has a weaker effect than water depth and calving front slope on the stability, as the magnitude and location of the stress maximum show a small sensitivity to variations in bed slipperiness.



A simple calving rate parametrization was found that predicts calving rates of tidewater glaciers in the Arctic reasonably well. This approach can be used to compute calving rates for grounded tidewater glaciers with relatively simple geometries, if ice thickness and water depth are known. The application of this parametrization in flow models of different complexity should be straightforward.

The present study lays the foundation for future, more detailed studies of the calving process on more realistic geometries.
Detailed analyses including time evolution, further processes such as frontal melt and water-filled crevasses, and data validation will be necessary for the implementation of improved calving parametrizations.

## Appendix A:  Stress parametrization

The distribution of longitudinal tensile stress at the surface $\widehat{\sigma}_{1s}$ can be fitted using stretched and scaled coordinates $\widehat{\widehat{x}}$ depending on relative water level $\omega$

$$\widehat{\widehat{x}} = 1.37\,\widehat{x} + 0.09 + \frac{0.031}{(1.07 - w)^2} \tag{A1}$$

The stress fit includes a taper towards the calving front which was chosen as an exponential. The full approximation to the stress curve is given by

$$\widehat{\sigma}_1(\widehat{x}) = a(w)\left(\widehat{\widehat{x}}\exp(-\widehat{\widehat{x}}) - \exp\left(\frac{-20\,\widehat{x}}{0.7 - \widehat{x}_m}\right)\right). \tag{A2}$$

The functions $a(w)$ and $\widehat{x}$ are given in Equations (17b) and (18).

*Acknowledgements.*  This work was funded by the Swiss National Science Foundation Grant 200021_156098



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





**Table 1.** Model parameters, notations, units and values for constant parameters.

| Parameter | Notation | Value | Units |
|---|---|---|---|
| Fluidity parameter | $A$ | 75 | $\mathrm{MPa}^{-3}\,\mathrm{a}^{-1}$ |
| Effective damage rate | $\tilde{B}$ | 37 | $\mathrm{MPa}^{-r}\,\mathrm{a}^{-1}$ |
| Bed slipperiness | $C$ | | $\mathrm{m\,MPa}^{-1}\,\mathrm{a}^{-1}$ |
| Initial damage | $D_0$ | 0.2 | |
| Critical damage | $D_c$ | 0.7 | |
| Gravitational acceleration | $\boldsymbol{g}$ | 9.81 | $\mathrm{m\,s}^{-2}$ |
| Sediment layer thickness | $h_s$ | 10 | m |
| Ice thickness | $H$ | | m |
| Water level height | $H_w$ | | m |
| Glen exponent | $n$ | 3 | |
| Damage law exponent | $r$ | 0.43 | |
| Velocity vector | $\boldsymbol{u}$ | | $\mathrm{m\,a}^{-1}$ |
| Basal velocity | $u_b$ | | $\mathrm{m\,a}^{-1}$ |
| Calving rate | $\bar{u}_c$ | | $\mathrm{m\,d}^{-1}$ |
| Reference velocity | $u_{\mathrm{ref}}$ | $\sim[1.7\cdot10^{-6}\mathrm{m}^{-3}\,\mathrm{a}^{-1}]\,H^4$ | $\mathrm{m\,a}^{-1}$ |
| Hayhurst parameter 1 | $\alpha$ | 0.21 | |
| Hayhurst parameter 2 | $\beta$ | 0.63 | |
| Hayhurst stress | $\chi_H$ | | MPa |
| Strain rate tensor | $\dot{\boldsymbol{\varepsilon}}$ | | $\mathrm{a}^{-1}$ |
| Effective strain rate | $\dot{\varepsilon}_e$ | | $\mathrm{a}^{-1}$ |
| Effective viscosity | $\eta$ | | $\mathrm{MPa\,a}$ |
| Sediment layer viscosity | $\eta_s$ | | $\mathrm{MPa\,a}$ |
| Finite strain rate parameter | $\kappa_\varepsilon$ | $5.98\cdot10^{-6}$ | $\mathrm{a}^{-1}$ |
| Ice density | $\rho_i$ | 917 | $\mathrm{kg\,m}^{-3}$ |
| Sea water density | $\rho_w$ | 1028 | $\mathrm{kg\,m}^{-3}$ |
| Fresh water density | $\rho_w$ | 1000 | $\mathrm{kg\,m}^{-3}$ |
| Cauchy stress tensor | $\boldsymbol{\sigma}$ | | MPa |
| von Mises stress | $\sigma_e$ | | MPa |
| Maximum principal stress | $\sigma_1$ | | MPa |
| Mean stress | $\sigma_m$ | | MPa |
| Deviatoric stress tensor | $\boldsymbol{\sigma}'$ | | MPa |
| Reference stress | $\sigma_{\mathrm{ref}}$ | $\rho_i g H \sim [0.009\,\mathrm{MPa\,m}^{-1}]\,H$ | MPa |
| Damage threshold stress | $\sigma_{\mathrm{th}}$ | 0.17 | MPa |
| Basal shear stress | $\tau_b$ | | MPa |
| Relative water level | $\omega$ | | |
| Relative water level at flotation | $\omega_f$ | 0.89 | |





**Table 2.** Maximum scaled Hayhurst stress and velocity for water depth experiments. The s and f letters indicate if the scaled Hayhurst stress maxima are found at the surface or at the bottom of the calving front, respectively.

| $\omega$ | $\max(\widehat{\chi}_H)$ | $\max(\widehat{u})$ |
|---|---|---|
| 0 | 0.808(f) | 1.098 |
| 0.25 | 0.513(f) | 0.967 |
| 0.50 | 0.323(s) | 0.526 |
| 0.75 | 0.193(s) | 0.097 |
| 0.85 | 0.123(s) | 0.019 |
| $\omega_f$ | 0.083(s) | 0.005 |

**Table 3.** Maximum Hayhurst stress and velocity for calving front slope experiments.

| Slope | $\max(\widehat{\chi}_H)$ | $\max(\widehat{u})$ |
|---|---|---|
| 90 | 0.808 | 1.098 |
| 75 | 0.752 | 0.722 |
| 60 | 0.669 | 0.465 |
| 45 | 0.571 | 0.269 |

**Table 4.** Maximum Hayhurst stress and velocity for bed slipperiness coefficient experiments.

| $C$ | $\max(\widehat{\chi}_H)$ | $\max(\widehat{u})$ |
|---|---|---|
| 0 | 0.323 | 0.526 |
| 333 | 0.328 | 0.579 |
| 666 | 0.333 | 0.634 |
| 1000 | 0.337 | 0.688 |





**Table 5.** Values of calving front height, water depth and calving rate for different glaciers.

| Abbr | Glacier | $H_s$ | $H_w$ | $u_c$ | Source |
|------|---------|-------|-------|-------|--------|
|      |         | (m)   | (m)   | $(\mathrm{m\,a^{-1}})$ |        |
| Bow  | Bowdoin 2015 | 25 | 220 | 550 | Sugiyama et al. (2015), pers. comm. G. Jouvet |
| Col  | Columbia 1983 | 53 | 213 | 2452 | Pfeffer (2007) Fig. 4 |
| Col  | Columbia 1988 | 29 | 243 | 3886 | Pfeffer (2007) Fig. 4 |
| Col  | Columbia 1994 | 61 | 280 | 5603 | Pfeffer (2007) Fig. 4 |
| Col  | Columbia 1998 | 103 | 253 | 10849 | Pfeffer (2007) Fig. 4 |
| Col  | Columbia 2000 | 122 | 260 | 9037 | Pfeffer (2007) Fig. 4 |
| Eqi  | Eqip Sermia 2015 | 50 | 80 | 3000 | Lüthi et al. (2016); Rignot et al. (2015) |
| Hel  | Helheim 2015 | 80 | 615 | 9125 | Murray et al. (2015); Voytenko et al. (2015) Fig. 2 |
| Hum  | Humboldt N 2015 | 25 | 250 | 450 | Carr et al. (2015) |
| Hum  | Humboldt S 2015 | 30 | 125 | 80 | Carr et al. (2015) |
| JI   | Jakobshavn Isbræ 2008 | 100 | 800 | 13000 | Lüthi et al. (2009) |
| Kng  | Kangilgata 1962/63 | 40 | 350 | 1650 | Carbonnell and Bauer (1968) |
| Lil  | Lille 1962/63 | 25 | 230 | 550 | Carbonnell and Bauer (1968) |
| Moe  | Moench 2006 | 50 | 0 | 35 | Pralong (2006) |
| RI   | Rink Isbræ 1962/63 | 70 | 560 | 4745 | Carbonnell and Bauer (1968) |
| Sto  | Store 1962/63 | 65 | 500 | 6300 | Carbonnell and Bauer (1968) |
| Sto  | Store 2015 | 60 | 500 | 5840 | Rignot et al. (2015); Ryan et al. (2015) |
| Yak  | Yakutat 2015 | 30 | 325 | 140 | Trüssel et al. (2015) |



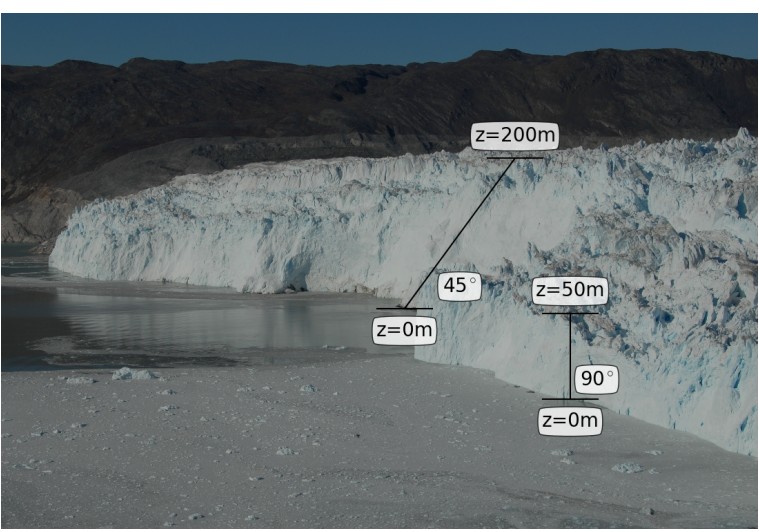

**Figure 1.** Calving front of Eqip Sermia glacier in July 2016. The boxes in the picture describe the geometrical properties of the two distinct parts of the calving front.

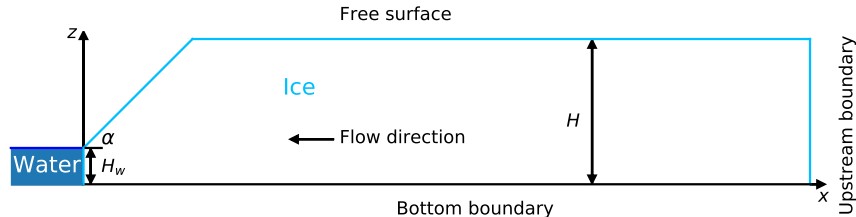

**Figure 2.** Geometry of the idealized grounded glacier. $\alpha$ is the slope angle of the calving front above the vertical cliff.



**Figure 3.** Sensitivity experiment results for varying water depth. A: Scaled Hayhurst stress distribution. B: Scaled horizontal velocity distributions. C: Scaled Hayhurst stress along the surface. D: Scaled horizontal velocity magnitude along the surface. In panels A and B, the subplots show increasing water depth from the bottom to the top (water level at $\hat{z} = 0$). Solid and dashed lines in Panels C and D correspond to experiments with sea and fresh water densities, respectively.



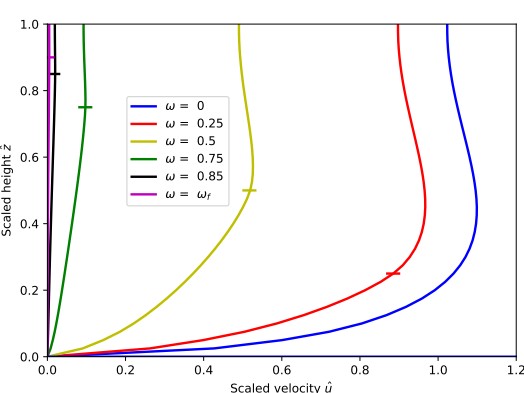

**Figure 4.** Scaled velocities along the vertical face of the calving front (solid lines) for different relative water levels. Horizontal line markers show the relative water level for each curve.





**Figure 5.** Sensitivity experiment results for varying calving front slope. Panels A, B, C and D are the same as Fig 3. In panels A and B, the subplots show decreasing calving front slopes from the bottom to the top. In panel C, the local minimum of stress close to the calving front is located where the front reaches its maximal height. In panel D, vertical lines on the curves for inclined fronts mark the distance at which the maximal surface height is reached.







**Figure 6.** Sensitivity experiment results for varying bed slipperiness $C$. Panels A, B, C and D are the same as Fig 3. In panels A and B, the subplots show increasing bed slipperiness from the bottom to the top. Units for bed slipperiness $C$ are $\mathrm{m\,MPa\,a^{-1}}$



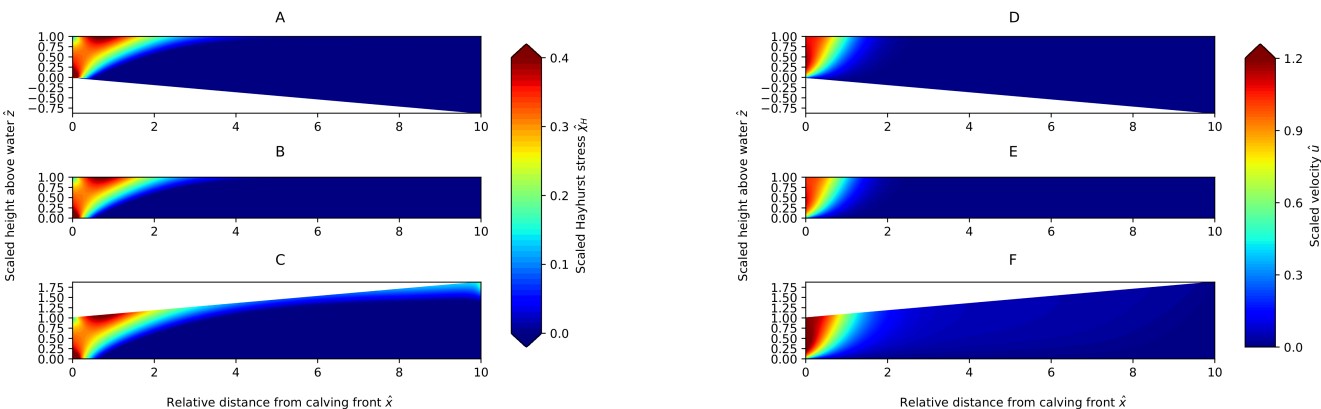

**Figure 7.** Sensitivity experiments result for an inclined surface (bottom), reverse bed (top) and the simple rectangular geometry (middle). The left and right panels show the scaled Hayhurst stress and velocity distributions, respectively.

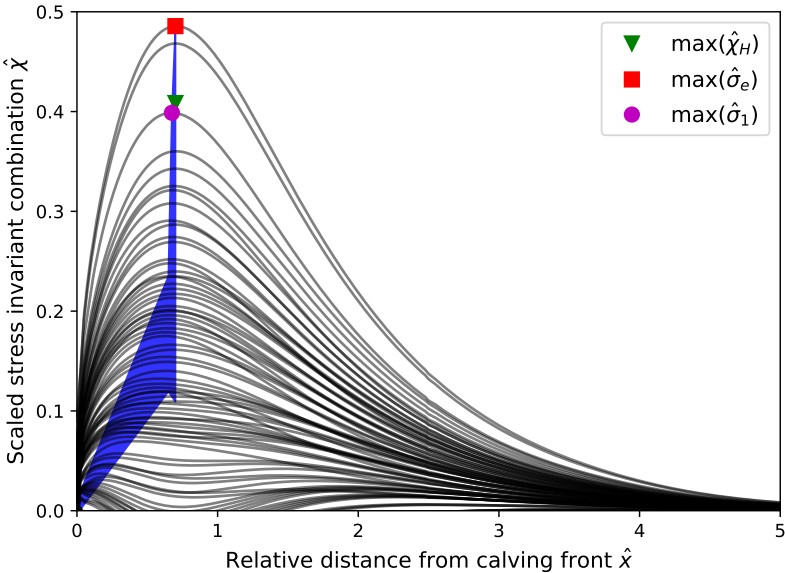

**Figure 8.** Combinations of five stress tensor invariants at the surface of an idealized glacier with a vertical calving front without water pressure and zero basal motion. Each black line represents a linear combination of five stress invariants. The blue envelope contains the maxima of all stress invariant combinations. The green triangle, red square and purple circle represent the maximum of the scaled Hayhurst stress $\widehat{\chi}_H$, von Mises stress $\widehat{\sigma}_e$, and maximum principal stress $\widehat{\sigma}_1$, respectively.





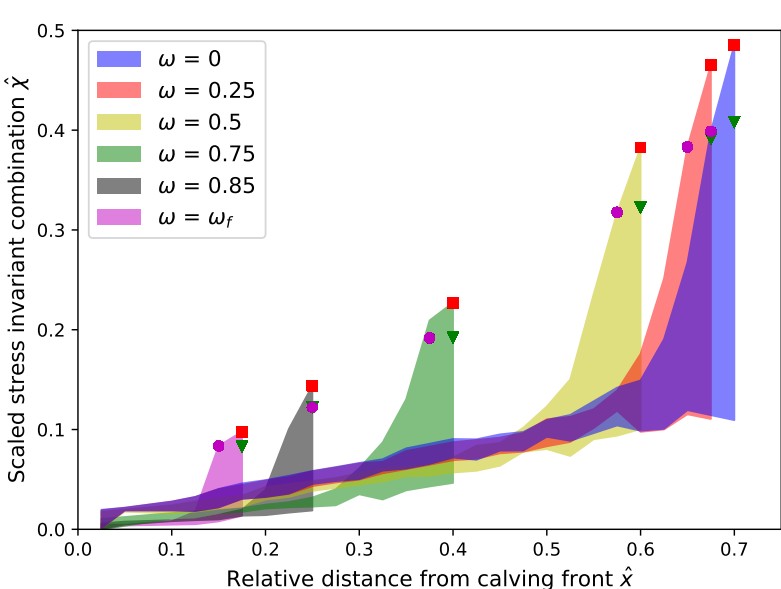

**Figure 9.** Envelopes of stress invariant combinations at the surface of the idealized glacier with zero basal motion for varying relative water level $\omega$. The green triangle, red square and purple circle represent the maximum of the scaled Hayhurst stress, von Mises stress and maximum principal stress, respectively, for each water depth.





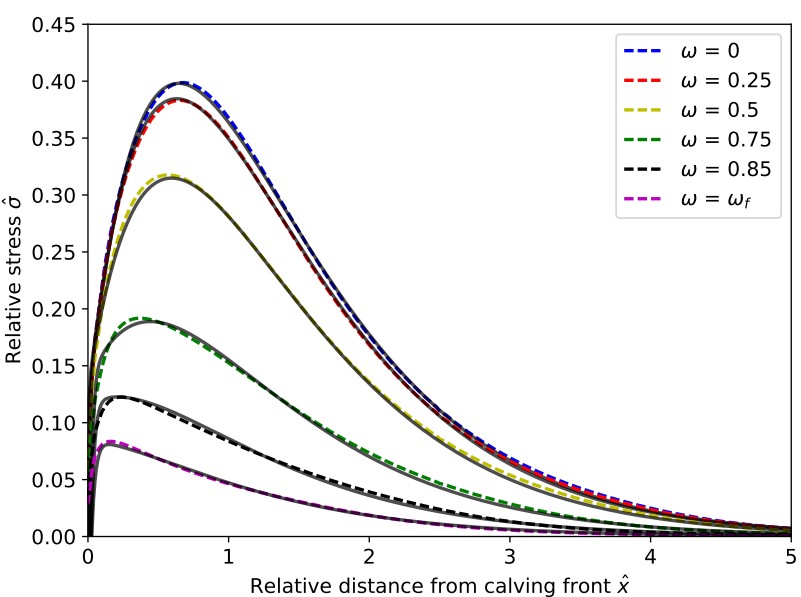

**Figure 10.** Modeled (dashed lines) and corresponding parametrized (solid lines) maximum extensive stresses $\widehat{\sigma}_1$ at the surface for different water depths. The dotted lines show the horizontal deviatoric stresses at the calving front for all water depths based on Eq.1




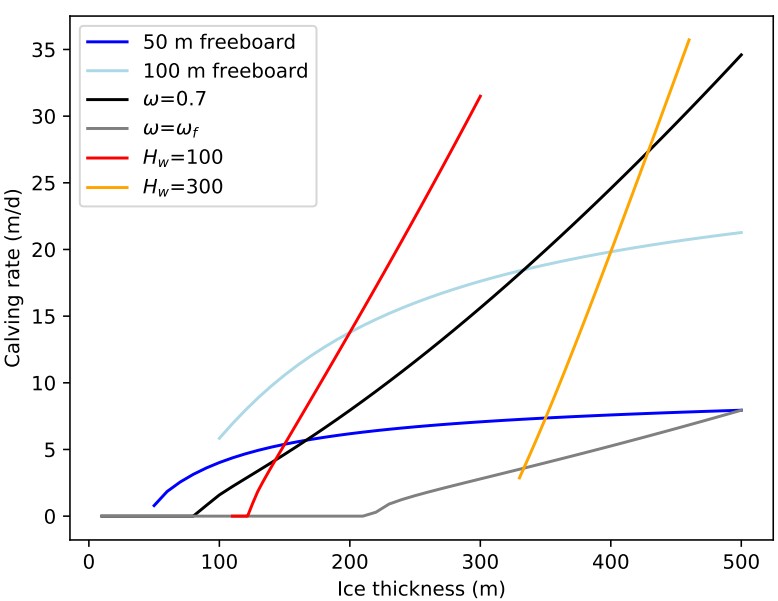

**Figure 11.** Calving rates predicted by the parametrization in relation to ice thickness. Calving rates increase with increasing total ice thickness for a given water depth $H_w = \omega H$ (red and orange lines), relative water level $\omega$ (black and gray lines) or freeboard $H - H_w$ (blue lines). Note that the gray line refers to a front at flotation.





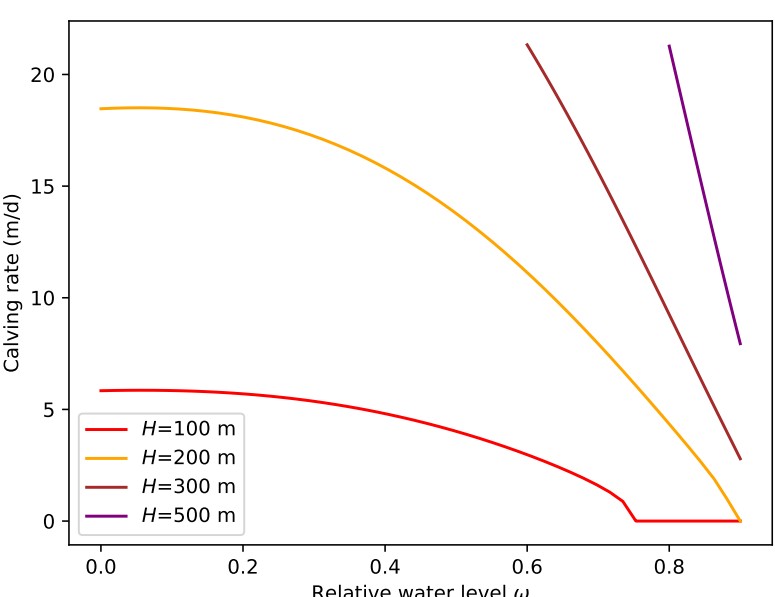

**Figure 12.** Calving rates predicted by the parametrization as a function of relative water level. Calving rate decreases under increase of the relative water level $\omega$ for constant total ice thickness $H$.



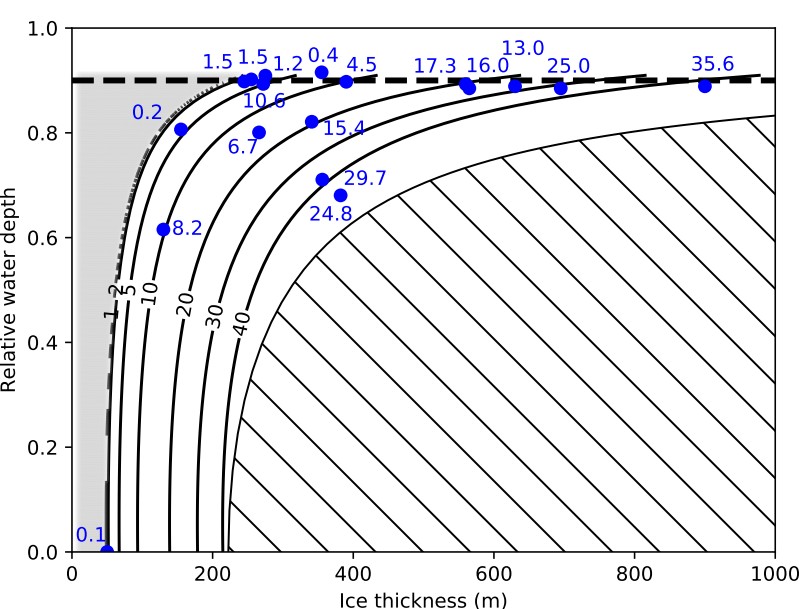

**Figure 13.** Calving rates $(\mathrm{m\,d^{-1}})$ predicted by the parametrization are shown as contours in dependence of $H$ and $\omega$. The hatched region indicates the states excluded by the maximum calving front criterion (Bassis and Walker, 2012). The gray area indicates states where the stress threshold $\sigma_{\mathrm{th}}$ precludes calving. Blue dots with numbers indicate calving rates determined from measurements, shown in Table 5.



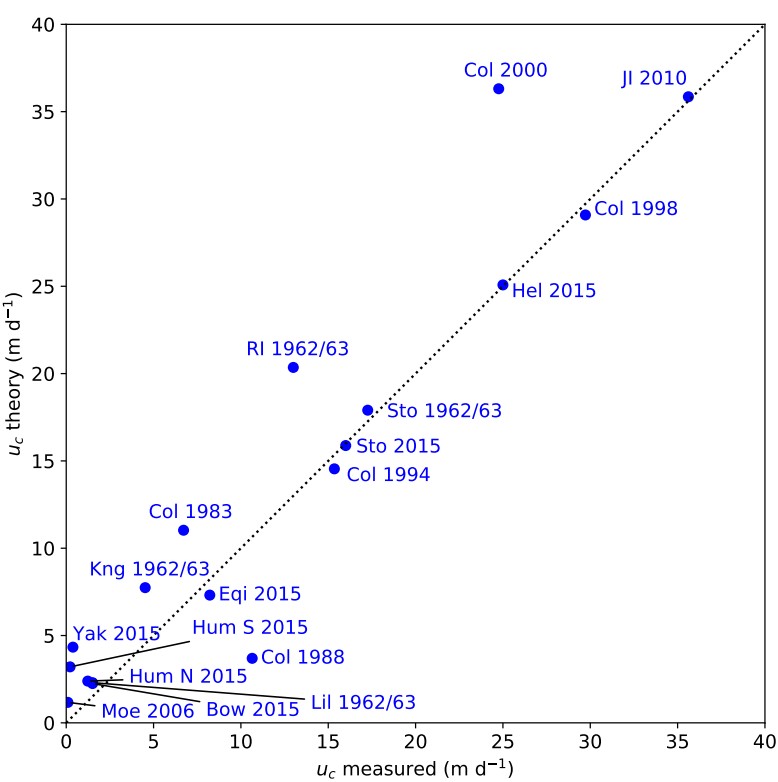

**Figure 14.** Comparison of measured calving rates with predictions from the calving parametrization. The glacier names are abbreviated according to Table 5.