# Peer review of "Calving relation for tidewater glaciers based on detailed stress field analysis"

_The Cryosphere, 2017_

## Short Comment (SC1) · 21 Sep 2017

D. Benn

dib2@st-andrews.ac.uk

The paper by Mercenier, Lüthi and Veili presents a new calving relation based on analyses of stresses and a damage evolution function. New approaches on the calving problem are welcome, and the paper contains some useful perspectives and comments. However, the paper suffers from some flaws. First, the calving relation is not entirely physically-based as claimed, but is actually semi-empirical and relies on some questionable assumptions. Second, the authors' claim that their model is "in good agreement with observations" is unjustified, because the model has simply been tuned to fit a set of observations, not validated against independent data. Third, there are significant inconsistencies between Eq. 22 and Figures 11,13 and 14, each of which produces a different calving rate for any chosen glacier in Table 5. Fourth, the authors

have misrepresented the existing literature on calving modelling, providing misleading context for their work.

Model formulation

The Hayhurst stress used in the first part of the analysis is not a physical quantity - it is an ad hoc combination of different stress metrics that - in the absence of any physical understanding of mechanisms - can be tuned to match observations. The Hayhurst stress is, in essence, a semi-empirical approach to divining which factors may control calving behaviour. This notwithstanding, the authors then abandon the Hayhurst stress approach and adopt the maximum principal stress as the foundation of their calving relation. The maximum principal stress is exactly the same stress metric used by Benn et al. (2017), and it is in fact a generalisation of the Nye crevasse depth formula used by Benn et al. (2007a). (Martin Lüthi provided a detailed review of Benn et al (2017), so at least one member of the author team has been aware of these results for some time.) The maximum principal stress is also equivalent, in 2D, to the 'effective stress' metric used by Todd & Christoffersen (2014) to model calving at Store Glacier.

The authors then formulate a calving rate law using a damage evolution function. Calving rate laws are attractive from a modelling perspective, but their physical justification is unclear. Rate laws are likely valid where calving is driven by melt undercutting, although this process is excluded from the analysis of Mercenier et al. In our work, we have focused on calving position laws, which predict the location of the calving front from the state of stress at any given time. Model experiments with the discrete element model HiDEM provide justification for this approach, because calving events occur rapidly in response to specific states of stress (Benn et al., 2017). Given the existence of these contrasting approaches to formulating calving laws, we feel that some discussion of this issue would benefit the present paper, and ideally the authors should provide more detailed justification for choosing a rate law.

In lines 317-8, the authors claim that their proposed calving parameterization is

"physics based, as opposed to the purely or semi-empirical nature of other approaches". This is wrong on two counts. First, crevasse depth calving laws (Benn et al., 2007; Nick et al., 2010; Todd and Christoffersen 2014; Benn et al. 2017) are physics based. Second, the calving parameterization proposed by Mercenier et al. is itself 'semi-empirical', and is reliant on tuning to data.

Model predictions and observations

In Section 5.3, the authors obtain values for two empirical parameters B (damage evolution rate) and sigma_th (damage threshold) using data from calving glaciers in the Arctic. The text in this section is rather obscure, but it seems that the data plotted in Figs. 13 and 14 are the same as those used for model tuning. Thus, the calving data in Fig 13 (wrongly described as 'velocity data' on line 340) are not shown 'for comparison', but are in fact the data points used to tune the position of the isolines of calving rate. Furthermore, Figure 14 does not compare calving data with model predictions, but compares calving data with predictions from a calving law tuned using the same data. It is therefore a representation of model fit rather than model performance. The authors are not justified in claiming that the model "predicts calving rates. . . reasonably well" (line 355), or that it is "in good agreement with observations" (line 10). The results simply mean that it is possible to tune the model to fit the data, not that the model has actual predictive power.

Errors in data plotting

Equation 22 and Figures 11, 13 and 14 do not seem to show the same calving parameterization. For example, plugging the "Columbia 2000" data into Eq. 22 (H = 382m, Hw = 260m, w = 0.68, measured Uc = 24.7 m d-1), with B = 37 MPa-r a-1, $\sigma$th = 0.17 MPa, r = 0.43, â■ťig = 0.009, gives a calving rate of 20.72 m d-1. However, in Figure 11, taking w $\sim$ 0.7, and ice thickness = 382m, calving rate is > 23 m d-1. In Figure 13, the intersection between H = 382m and w = 0.67 gives a value greater than 45 m d-1. Finally, in Figure 14, 'Col 2000' is shown with a predicted calving rate of $\sim$36 m d-1.

Review of previous literature

Previous literature is cited in a very partial way, and some previous work is misrepresented. For example (line 34) the authors state that Benn et al. (2007a, b) "generalized the flotation criterion", and later (line 318) they imply that these papers take an empirical or semi-empirical approach to modelling calving. These statements are untrue. The papers by Benn et al. proposed a new, physically based approach to modelling calving, setting the position of the calving front where crevasses penetrate to the waterline (Benn et al., 2007a). This was later modified by Nick et al. (2010) to include crevasse penetration through the full thickness of the glacier. These initial formulations computed crevasse depths from only longitudinal stresses, but subsequent work has generalised the crevasse criterion to include extensional stress in 2D (Todd and Christoffersen, 2014) and 3D (Benn et al., 2017). Benn et al. (2017) also discussed at length the issue of stress balance "snapshots", and proposed strategies for overcoming these limitations. The criticism that crevasse depth models lack "validation with field observations" (line 39) is also unwarranted. Some authors have tuned the model to match observations (e.g. Nick et al., 2014; Lea et al., 2014), which is exactly the same approach as taken by Mercenier et al. Comparison of the predictions of an untuned crevasse depth model against independent observations has been done by Todd et al. (in press), although of course Mercenier et al. cannot be expected to cite this work. A copy can be supplied on request.

In lines 42-50, the authors present an approximation for depth averaged longitudinal stress and state that it is the "main driving force" of crevasse depth models. However, the crevasse depth criterion is not fundamentally a depth-averaged law, although it has previously been implemented in 1D dynamic models (e.g. Nick et al. 2010). Todd and Christoffersen (2014) implemented the crevasse depth criterion in a 2D full-Stokes model, similar to that presented by Mercenier et al., and computed crevasse penetration locally based on nodal stresses. The version of the crevasse-depth calving law adopted by Benn et al. (2017) uses the maximum principal stress, which is exactly the

same metric used by Mercenier et al. in the second part of their analysis.

The review of melt undercutting (lines 63-67) also misrepresents the literature. Hanson and Hooke (2000) did not look at undercutting at all, and O'Leary and Christoffersen did not "suggest that an increase of water depth leads to a higher rate of oversteepening development". In fact, O'Leary and Christoffersen argued that increasing water depth increased the stress response to a given amount of undercutting, and hence the magnitude of the 'calving multiplier' effect. The opposite trend was found in the more detailed model experiments by Benn et al. (2017), who showed that the effect of undercutting is diminished by increasing water depth. Cook et al (2014) and Krug et al (2015) did indeed conclude that melt undercutting does not significantly affect calving rates. However, this is due to the insensitivity of their models to undercutting, as discussed in detail by Benn et al. (2017). In fact, recent studies have shown that melt undercutting is the primary driver of calving in Svalbard and some Greenland fjords (Luckman et al., 2015; Cowton et al., 2016).

Additional Point

Regarding the role of basal sliding, the authors provide a useful perspective on their results in lines 303-4, in which they point out that spatial variations in basal slipperiness (as would result from a pressure-dependent sliding law) would likely introduce velocity gradients that could affect calving. This caveat is not reflected in the statements in the abstract ("the effect from VARIATIONS in basal sliding is much smaller" (emphasis added) and the conclusions ("basal sliding likely has a weaker effect ...on stability": lines 353-4). Because they only impose uniform basal slipperiness, the experiments presented in this paper cannot evaluate the influence of basal sliding on calving - and this includes their important relationship with water depth. The summary statements in the abstract and conclusions should reflect this.

Concluding remarks

The proposed calving relation has the benefit of simplicity, and makes some interesting

and testable predictions. However, we are sceptical this it represents an improvement on existing approaches because it is based on only one control on calving - the stresses introduced by the force imbalance at the ice front - and neglects other important processes. Calving also occurs in response to longitudinal stresses caused by along-flow variations in basal and lateral drag; melt undercutting in response to heat flux from the ocean; and super-buoyancy where glaciers flow rapidly into deep water. Indeed, these processes are known to be the main drivers of calving on several of the glaciers listed in Table 5. The predictions of the model listed in lines 321-326 correspond to some observed behaviour (e.g. the increasing instability of high, unsupported ice cliffs) but not others (e.g. calving triggered by ice flow into deepening water).

But of course the search for a general calving law goes on, and alternative approaches may help us reach this elusive goal. With greater awareness of its limitations and better perspective on its context within the wider literature, the paper by Mercenier et al could provide a useful contribution.

Doug Benn and Joe Todd

Additional references

Benn, D.I., Åström, J., Zwinger, T., Todd, J., Nick, F.M., Hulton, N.R.J. and Luckman, A. 2017. Melt-undercutting and buoyancy-driven calving from tidewater glaciers: new insights from discrete element and continuum model simulations. Journal of Glaciology 63 (240), 691-702.

Cowton, T., Sole, A., Nienow, P., Slater, D., Wilton, D. and Hanna, E., 2016. Controls on the transport of oceanic heat to Kangerdlugssuaq Glacier, East Greenland. Journal of Glaciology 62, 1167-1180.

Lea, J.M., Mair, D.W., Nick, F.M., Rea, B.R., Weidick, A., Kjær, K.H., Morlighem, M., Van As, D. and Schofield, J.E., 2014. Terminus-driven retreat of a major southwest Greenland tidewater glacier during the early 19th century: insights from glacier reconstructions and numerical modelling. Journal of Glaciology, 60(220), pp.333-344.

Todd, J. and Christoffersen, P.: Are seasonal calving dynamics forced by buttressing from ice mélange or undercutting by melting? Outcomes from full-Stokes simulations of Store Glacier, West Greenland, The Cryosphere, 8, 2353-2365, https://doi.org/10.5194/tc-8-2353-2014, 2014.
* * *

---

## Referee Comment (RC1) · J. Otero (Referee) · 1 Oct 2017

This manuscript investigates the influence of water level, calving front slope and basal sliding on the stress and flow regimes in the vicinity of the calving front and proposes a new parameterization for calving. This relation has the advantage of its simplicity and ease of use, since it is only function of ice thickness and the water depth at the calving front. However, the manuscript has some issues that I think should be addressed prior to publication.

In the Introduction section, when the authors review the previous literature on calving modeling some previous work is misrepresented and the overall perspective is confused. I think that some paragraphs, at least from line 34 to line 71, have to be

rewritten, keeping in mind the following tips.

- Lines 34-35. The crevasse depth criterion (Benn et al, 2007) isn't a generalization of the flotation criterion, is a completely different approach to the calving problem based on the physics.

- In lines 42-50 it seems that the authors claim that the depth averaged longitudinal stress is the "main driving force" of crevasse-depth calving based models. But from Otero et al. (2010) such limitation was overcome and subsequent models (e.g. Cook et al., 2014; Todd and Christoffersen, 2014; Otero et al., 2017; Benn et al., 2017) computed crevasse penetration locally based on nodal stresses.

- I also note the lack of any reference of the recent work of Benn et al. (2017) in which crevasse-depth calving approach they uses the maximum principal stress, which is the same metric used by the authors in the second part of the paper.

- The importance of melt undercutting in the process of calving is underrepresented in lines 63-71. In some modeling works cited in the manuscript (Cook et al., 2014; Krug et al., 2015) did indeed conclude that melt undercutting does not significantly affect calving rates. On the contrary, in recent studies has been shown that melt undercutting play an important role in calving of some Arctic glaciers (Luckman et al., 2015; Petlicki et al., 2015; Cowton et al., 2016).

In lines 317-318 I miss any reference to the crevasse-depth calving criterion, that is a physics-based approach. And the calving parameterization proposed by the authors is "semi-empirical" since they tune two parameters using observational data.

In section 5.3 the authors describe how they tune the model against data from some tidewater glaciers in the Arctic, so I recommend to rename this section. And the statement "In this study we used the values shown in Table 5 from diverse data sources for comparison" (line 338) is unclear. The authors are using the data showed in Table 5 to tune the parameters of the model, not for comparison.

[Figure]

It seems that, in the last two paragraphs of section 5.3, the authors compare their model predictions with the same data used to tune the model. So the statements "the general agreement shows that the parameterization is suitable to predict calving rates for many tidewater glaciers in the Arctic" (lines 344-345), "A simple calving rate parameterization was found that predicts calving rates of tidewater glaciers in the Arctic reasonably well" (line 355) and "we propose a simple and new parameterization for calving rates for grounded tidewater glaciers that is in good agreement with observations" (line 10) are not fully justified.

Check on the errors in data plotting pointed out by Benn and Todd in their Short Comment.

340. "Calving rates" instead "Velocity data"

344. "rates" instead "velocities"

Change the units of calving rate in Table 5 to md-1 in coherence with Figures 13 and 14.

---

## Referee Comment (RC2) · J. Bassis (Referee) · 3 Oct 2017

**General Appreciation**

This paper analyzes the stress near the calving front of idealized glaciers. Then, by relating the state of stress, as measured by the Hayhurst criterion, to different ice thicknesses and water depths, the authors combine the stress estimates with an analytic model of damage mechanics to estimate calving rates. The model proposed only depends on three parameters, but these parameters must be determined by comparison with observations, field or laboratory. In this case, the model is calibrated to a suite of Arctic glaciers and the model is shown to be reproduce observed calving rates for the glaciers for which the model is calibrated.

[Figure]

The paper takes an impressively innovative approach to an old problem and the agreement between observations and model predictions is impressive, although it is hard to evaluate the models performance given the fact that it was tuned to match this specific set of glaciers. I suspect that the approach pioneered in this paper will ultimately become more commonly used. I do, however, have a few significant questions about the physical interpretation of the mathematical formulation. This paper has already benefited from the insightful comments of two very well qualified reviewers. My comments both build and diverge from these reviewer comments so I will first wade into the discussion already initiated by the reviewers and try to cast it in a slightly different perspective. This will hopefully better motivate my comments about the physical interpretation of the calving law that I present as the last point in the major comments section.

**Major comments**

1. *What is the right stress metric to use and why?* The authors of this study use the Hayhurst criterion, a linear combination of stress tensor invariants. In contrast, Doug Benn and Joe Todd argue that this is a purely empirical relationship and that the largest principle stress is the more physics based metric and results from a generalization of Nye's zero tensile stress model. Physics tells us that in the absence of anisotropy, the failure criterion should depend on invariants of the stress tensor. However, it is unclear to me how or why *physics* provides any guidance as to which invariants it depends on. To be more clear, in the uniaxial case, failure is clearly related to the single component of stress and because ice, like most materials, is much weaker in tension than compression observations show that (tensile) failure occurs when the applied strength exceeds some threshold. In the Nye zero stress model, that threshold is set to zero. We know from experiments and field observations that ice has finite strength. Fortunately, Weertman showed that for fields of closely spaced crevasses, the depth of crevasses will approximate the Nye zero stress depth so long as the initial starter crack length

is sufficiently large and the strength of ice, measured as critical stress intensity factor, sufficiently small. The generalization to multi-axial failure is less obvious. Benn et al., (2017) apply the maximum principle stress. We did the same thing in Yue and Bassis, (2017), but also considered the possibility that shear failure could occur. Under multi-axial loading there are a larger number of invariants of the stress tensor that must be considered. The Hayhurst criterion attempts to combine multiple modes of failure in a single fracture growth model by taking a linear combination of invariants. Both the Hayhurst and maximum principle stress criterion reduce to the uniaxial case when other stress components vanish and one could view the Hayhurst criterion as a generalization of the Nye criterion to multi-axial loading. The distinction between the two hypotheses is that the maximum principle stress criterion predicts that multi-axial loading will have *no* effect on the depth or rate of fracture propagation. To interpret the maximum principle stress criterion rigidly, implies that failure of ice only occurs through tensile failure and no other mode of failure is possible, an assertion that is falsified by laboratory measurements. In contrast, the Hayhurst criterion tells us that multi axial loading can increase the depth, trajectory and rate of fracture propagation. Moreover, the Hayhurst criterion reduces to the maximum principle stress criterion and is thus more general. Crucially, I don't see a way to deduce which— if either— is correct in the absence of observations. It is, however, clear that the maximum principle stress criterion has provided useful results that allow us to predict how fields of crevasses respond to changes in stress. That this relationship is not deduced from a more fundamental principle doesn't detract from its usefulness. It is also entirely possible that assuming under glaciological regimes tensile failure dominates is a useful *approximation*. I'm uncomfortable litigating the physical appropriateness of one model versus another without observational data to confirm or refute hypotheses, which is something the authors might think about reviewing. *Where I think this manuscript could improve is to provide a better motivation for why the Hayhurst stress is first introduced and then, why it*

*is abandoned in favor of the maximum principle stress. I think a valuable result would be to show that results are insensitive to the choice of failure metric (Hayhurst or maximum principle stress). We may not know which is the correct law, but it might not matter. Similarly, it would be helpful if the authors could comment on any observational evidence to support or refute the use of the Hayhurst stress in glaciological applications.*

2. *Calving rate versus calving positions laws:* As a minor point, the relationship between calving rate and calving position laws is in fact direct (see Bassis, 2010). Calving rate laws can be deduced from statistical averages over many calving events and are valid over time scales that are much longer than the typical recurrence interval between calving events provided the spatial scale of calving events is small compared to the glacier system. The relationship between the two, in a statistical sense, is really just a switch in time scale. Calving position laws have the advantage that they better encompass fluctuations in calving front position, but are less practical over longer timescales (e.g., millennial) when the only data available is average position at discrete intervals of time. Moreover, many of the 'position laws' can be equivalently formulated as rate laws. A simple example of this is the height-above-buyancy law and others in its family. It is straightforward to cast this as a continuous rate law for calving front position analogous to those used for grounding line migration. *What is a more fundamental issue for me is whether the terminus position of the glaciers used to calibrate the law are relatively constant or changing. If the glaciers are in or near steady-state than many variables can be correlated without indicating causality. A more convincing argument is if the calving law can predict the rate of retreat/advance for one or more glaciers that is changing.*

3. *Empirical, semi-empirical versus physical calving laws:* Both reviewers brought up the point that the calving law is empirical rather than physical. I have similar concerns, although my physical concerns are slightly different and will be raised

in the point that follows. To preface, I do increasingly worry that we are using the concept of 'physical' versus empirical as a blunt cudgel to beat each other. The rheology of ice that we use is an empirical flow law. The formulation presented by Cuffey and Paterson is excellent and results from a set of calibrated model experiments and laboratory measurements. Despite this empirical basis, most of us don't usually describe our ice dynamics models as 'empirical' or 'semi-empirical' due to the fact that the parameters in Glen's flow law are not calculated from first principles. We glaciologists scorn well calibrated empirical data at our peril. *Instead of wading into the empirical debate, I would encourage the authors to ask a couple of questions. (1) What predictions can the calibrated model make that are independent of the data set used to calibrate the model that can be used to falsify the model? The model is calibrated for a suite of Arctic glaciers, but can it be used to predict the calving rates of glaciers that are not included in the data set? For example, there are only a couple of datapoint for Columbia Glacier, but the retreat has been well documented for several decades. Alternatively, one could use different subsets of data to calibrate the model and then validate against an independent set. More physically, the authors are inferring a threshold stress at which damage begins to grow. This threshold could be compared to observations that indicate the stress at which surface crevasses first appear. (2) How sensitive are the results to the model calibrated parameters? If the model results only weakly depend on the calibrated parameters, then the fact that they are determined empirically is not much of a concern because we only need to get ballpark estimates. However, if the results depend sensitively on one or more parameters than we need to think careful about how to measure these parameters independently and may be concerned that the model predictions may be less reliable when applied elsewhere.*

4. *Physical interpretation of the calving law:* The point that I struggle with the most in this paper is physically interpreting the mathematical model. My interpretation

of Equation 22 is that authors are stuffing the maximum principle stress as measured at the surface into their damage evolution law and then evaluating how long it takes for a surface crevasse at the location of the maximum stress to develop. The calving rate is then the distance to the maximum stress at the surface divided by the time scale of the calving event. This calculation, however, seems to give the time scale for a crevasse at the surface to develop and **not** the time scale for a crevasse to penetrate the entire ice thickness or some fraction thereof. In simulations that we have done using a similar formulation of damage mechanics as presented here, but simulating the propagation of individual crevasses, dry surface crevasses never penetrate the entire ice thickness. I can accept the arguments that lead to a shallow surface crevasse at the surface, but the magic that then asserts that the surface crevasse will penetrate deep enough to cause a calving event is not clear. In fact, looking at the stress field, it looks like the maximum principle (or Hayhurst stress) decreases with depth. Why then does the surface crevasse propagate the entire distance and why doesn't it take longer to propagate down as the stress decreases? It is this step more than the details of the calibration that makes the calving law seem empirical or divorced from 'physics' to me. Getting fractures to propagate the entire ice thickness has always been a problem for calving models and this study seems to sidestep these issues. I guess I'm OK with postulating a calving form base on the position and magnitude of the surface tensile stress, but the departure from physics should be more clearly emphasized. Furthermore, given this departure, I don't quite understand why the complex damage evolution law is used. What if a linear relationship between damage and principle stress was postulated instead? The rate of damage growth would then have two parameters (a rate factor and stress threshold). Or what if the stress threshold was set to zero, giving a single parameter? Do these choices significantly degrade the fit? Does the calving law depend sensitively on the form of the assumed damage law or is this calibrated out? Can we say anything about the form that the damage evolution law must take if the data is to

be matched?

**Detailed comments**

Page 1 abstract "stress state" or "state of stress"?

Use of crack: considering using crevasse or defining how the term crack includes more than just crevasses for your glaciological audience.

Page 2, near line 35: "Benn et al. (2007a, b) generalized the flotation criterion by setting the terminus position at the location where crevasses penetrate below the water level" I think the authors are getting at the fact that in the height-above-buoyancy criterion, the position of the calving front is specified rather than a calving rate (as in the Brown et al., water depth model). The Benn et al., approach thus provides a different model to compute the terminus position based on when surface crevasses penetrate to the water line. This is mechanically different than the height-above-buoyancy criterion, but falls into a similar type of law whereby the position of the calving front is determined.

Page near line 40: "However, the crevasse depth estimation lacks validation with field observations and is based on a snapshot of the stress balance, neglecting the pre-existence of cracks and their effect on the stress state of the glacier (Krug et al., 2014)." This is an excellent point. Most crevasse penetration models assume that crevasses have a negligible effect on the state of the stress and are purely passive. These crevasse penetration models also ignore the advection of previous existing crevasses into the near terminus region. It is unclear how these effects are incorporated into the proposed model.

Equation (1) is a depth averaged equation. We don't have to rely on it and can instead compute approximations of the state-of-stress using finite element models. However, Equation (1) has the advantage that it is non-parametric (i.e., independent of ice rheology). More sophisticated methods of calculating the stress field require additional, often unknown parameters like the temperature of the ice and an appropriate sliding

law.

Page 2 near line 50: "The meaning of such a 'depth averaged' longitudinal stress for local fracture, for example for assessing surface crevasse formation, and the calving processes is not clear." I'm not sure I understand the complaint here. The depth integrated approach yields an estimate of the tensile stress based solely on the ice thickness and water depth. This estimate of stress allows us to estimate the depth of crevasses. Of course, near the terminus the depth integrated formulation may not accurately estimate the stress due to the absence of bending effects and neglected terms in the force balance. This seems like it points to a lack of accuracy rather than a difficulty with interpretation of the meaning of depth averaged stress.

Page near line 55: "However, observations of the appearance of surface crevasses on glaciers in relation to the strain rate field suggest a much lower cohesive strength of glacier ice between 0.09 and 0.32 MPa (Vaughan, 1993) for cold Antarctic ice streams, or as low as 0.05 MPa for a temperate Alpine glacier (Lliboutry, 2002)." This is an interesting point about the uncertainty in the yield strength, but unless I am misunderstanding, might be related to some confusion about different modes of failure. The fact that surface crevasses are detected at low stresses doesn't imply that the strength of ice in shear must be much lower. Tensile and shear failure can be distinct modes of failure and each can have their own yield strength.

Page 3 near line 70 "This implies that thick glaciers approach flotation at their front but for shallow water depth the constraints on geometry seem less clear." It is true that the bounds are very wide for small ice thicknesses. I would say that bounds are clear, but not particularly useful for small ice thicknesses given the spread of permissible values.

Equation 3: Should have a dot between the del and u to enforce the divergence of velocity is zero not the gradient of velocity is zero.

Numerical implementation: How is the incompressibility condition enforced numerically? Is this a mixed element formulation?

Section 2.2. This might be more clear if one uses the scales introduces to non-dimensionalize the governing equations. Then, I assume, we could write the equations in terms of a set of non-dimensional numbers, like the aspect ration (H/L) that describe the dynamic and geometric similarity between solutions.

Section 2.3. I think the boundary condition is traction free not stress free. One is not usually able to prescribe the entire stress tensor.

Page 5, line 130: It is also possible to specify zero slip in the vertical direction along the inflow boundary condition. This prevents edge effects near the zero velocity boundary condition.

Results: Section 3.1: It looks like when the authors say "stress" they mean Hayhurst stress, but what parameters were used to calculate the Hayhurst stress? I would have thought that the patterns of, say Von Mises stress would be very different from the largest principle stress? I can't find this information in the text or figure caption. This difficulty in understanding how the Hayhurst stress was calculated continues throughout the rest of the results section. Would it be more more helpful to show the largest principle stress, Von Mises stress in separate panels?

Page 7, line 185: The extrusion flow is an example of the Poison Effect.

Page 8, section 3.1.3. I'm not sure I understand the basal slipperiness results. When we did experiments using a full Stokes model we considered no-slip boundary conditions and free-slip boundary conditions and these two experiments resulted in significant differences in velocity and stress.

Page 9 line 235: "Figure 9 clearly illustrates that water pressure at the calving front exerts a stabilizing effect on the calving front by both lowering the stresses and decreasing the distance from the calving front at which the stress maximum is located." This is exactly what we argued in Bassis and Walker (2012), although our analysis was less numerically sophisticated.

Page 9, line 240: Now I think I have lost the thread. Why consider the Hayhurst criterion at all if the maximum principle stress is all that is going to be used? Is the reason the Hayhurst stress is going to be abandoned because all of the invariant combinations give similar answer? This could use a bit more motivation.

References:

Bassis, J.N.. (2010). The statistical physics of iceberg calving and the emergence of universal calving laws. Journal of Glaciology. 57. 10.3189/002214311795306745.

Ma, Y., C. S. Tripathy, and J. N. Bassis (2017), Bounds on the calving cliff height of marine terminating glaciers, Geophys. Res. Lett., 44, 1369–1375, doi:10.1002/2016GL071560.

---

## Author Comment (AC1) · 29 Nov 2017

We would like to thank the referees Jaime Otero and Jeremy Bassis, as well as Doug Benn and Joe Todd for their helpful comments on this manuscript. Below, all points raised (in black) are addressed (with responses in blue) and changes to the text are *emphasized* after each point. Changes to the manuscript are written in **red** in the new version.

**General points and response**

First, we respond to the main points raised by all commenting authors.

- Semi-empirical calving relation:

  We agree with the reviewers that the calving relation is semi-empirical and not entirely physics-based. This statement was changed in the new version of the manuscript.

- Comparison with observations:

  The data set was indeed used for calibration rather than comparison with data, with the figures showing the fit between the parametrization and the observations. Section 5.3 has been renamed "*Calibration of the parametrization*" and the wording has been adapted for all cases mentioned in the comments.

- Errors in data plotting:

  We would like to thank Joe Todd and Doug Benn for pointing out the errors of plotting. The figures (11,12,13,14) did indeed not show the same parametrizations. Figures 11, 12 and 13 were produced with a different value of $B$ than the calibrated one. The figures have been replaced using the calibrated parameters in the new version of the manuscript.

- Review of previous literature:

  We agree that the review of the literature was not fully complete and clear in the older version of the manuscript. The sections containing literature reviews have been adapted in the new version of the manuscript.

- Use of first principal stress instead of Hayhurst:

  We clarified why we used the first principal stress (instead of the Hayhurst stress) for the calving parametrization and add a new figure in the appendix of new version of the manuscript. The figure shows the stress distributions for Hayhurst and first principal stress for comparison.

- Calving rate vs. calving position relations:

  In this work, we do not argue for or against any type of calving relation. We believe that different calving relations can be useful for different purposes. We made clearer, however, that our new parametrization also implicitly (through ice thickness and water depth) includes the upstream dynamics for estimating the calving rate.

In the following, we address in detail all comments made by the reviewers.

**Referee comment by Jaime Otero (RC1)**

- Lines 34-35. The crevasse depth criterion (Benn et al, 2007) isn't a generalization of the flotation criterion, is a completely different approach to the calving problem based on the physics.

We agree that this was not well formulated, and clarified this point in the new version of the manuscript (lines 35-36):

> *Benn et al. (2007a,b) introduced a physics-based approach by setting the terminus position at the location where crevasses penetrate below the water level.*

- In lines 42-50 it seems that the authors claim that the depth averaged longitudinal stress is the "main driving force" of crevasse-depth calving based models. But from Otero et al. (2010) such limitation was overcome and subsequent models (e.g. Cook et al., 2014; Todd and Christoffersen, 2014; Otero et al., 2017; Benn et al., 2017) computed crevasse penetration locally based on nodal stresses.

This statement applies generally to steep calving fronts and is not related to any model. We make this distinction clearer by reformulating it. (lines 46-47):

> *For near-vertical calving fronts, the main driver for calving is the horizontal deviatoric stress $\sigma'_{xx}$ in vicinity of the laterally-confined calving front.*

- I also note the lack of any reference of the recent work of Benn et al. (2017) in which crevasse-depth calving approach they use the maximum principal stress, which is the same metric used by the authors in the second part of the paper.

The citation of Benn et al. (2017), which was not yet published when the manuscript was written, is now included in the new version of the manuscript (lines 42-45):

> *A recent, more sophisticated approach by Benn et al. (2017) predicts calving positions based on the maximum principal stress distribution, and accounts for the effect of water pressure in the submerged parts of the glacier front by combination of a continuum flow model with a discrete element model to simulate calving events*

- The importance of melt undercutting in the process of calving is underrepresented in lines 63-71. In some modeling works cited in the manuscript (Cook et al., 2014; Krug et al., 2015) did indeed conclude that melt undercutting does not significantly affect calving rates. On the contrary, in recent studies has been shown that melt undercutting play an important role in calving of some Arctic glaciers (Luckman et al., 2015; Petlicki et al., 2015; Cowton et al., 2016).

We provide now a more nuanced discussion of the role of undercutting on calving and included the additional references to melt undercutting from Cowton, 2016, Luckman 2015, Petlicki 2015, Benn 2017 (lines 65-71):

> *Calving termini can also be over-steepened by melt undercutting, which leads to higher stress intensities (Hanson and Hooke, 2000) and may facilitate calving (Benn et al., 2017). Ice flow model results (Hanson and Hooke, 2000) suggest that an increase of water depth leads to a higher rate of over-steepening development at the calving front and thus an increase of calving activity. However, model results seem to indicate that melt undercutting does not significantly affect calving rates (Cook et al., 2014; Krug et al., 2015) while other studies suggest that calving rates are strongly related to melt undercutting for some arctic glaciers (Luckman et al., 2015; Petlicki et al., 2015; Cowton et al., 2016). Conversely, a calving front inclined towards the inland is expected to be more stable than a vertical cliff.*

- In lines 317-318 I miss any reference to the crevasse-depth calving criterion, that is a physics-based approach. And the calving parametrization proposed by the authors is "semi-empirical" since they tune two parameters using observational data.

The reference to the crevasse-depth models has been added (lines 340-342). We agree that the calving parametrization is semi-empirical and clarified this point in the text.

> *The semi-empirical calving rate parametrization is therefore, in the sense of inclusion of upstream dynamics, similar to the position based calving models (Benn et al., 2007a; Nick et al., 2010; Todd and Christoffersen, 2014; Benn et al., 2017)*

- In section 5.3 the authors describe how they tune the model against data from some tidewater glaciers in the Arctic, so I recommend to rename this section. And the statement "In this study we used the values shown in Table 5 from diverse data sources for comparison" (line 338) is unclear. The authors are using the data showed in Table 5 to tune the parameters of the model, not for comparison.

This is correct, we adjusted the wording and clarified that we calibrated the model with this data. We also renamed the section title to "*Calibration of the parametrization*" and the term *comparison* has been replaced by *calibration*.

- It seems that, in the last two paragraphs of section 5.3, the authors compare their model predictions with the same data used to tune the model. So the statements "the general agreement shows that the parametrization is suitable to predict calving rates for many tidewater glaciers in the Arctic" (lines 344-345), "A simple calving rate parametrization was found that predicts calving rates of tidewater glaciers in the Arctic reasonably well" (line 355) and "we propose a simple and new parametrization for calving rates for grounded tidewater glaciers that is in good agreement with observations" (line 10) are not fully justified.

We agree and these statements have been adapted as follows

Lines 358-359:

> *While a sizable spread of the data is visible, especially for low calving rates, the general agreement shows that the parametrization is well suited to estimate calving rates for this set of tidewater glaciers in the Arctic.*

Lines 372-373:

> *A simple calving rate parametrization was derived that was calibrated with calving rate data of a set of tidewater glaciers in the Arctic*

Lines 8-10:

> *Based on this scaled relationship for the stress peak at the surface, and assuming a critical stress for damage initiation, we propose a simple and new parametrization for calving rates for grounded tidewater glaciers that is calibrated with observations.*

Check on the errors in data plotting pointed out by Benn and Todd in their Short Comment.

The errors in data plotting have been corrected in the new version of the manuscript. The figures did not show the same parametrizations. Figures 11, 12 and 13 were produced with a different value of $\tilde{B}$ than the calibrated one.

- 340. "Calving rates" instead "Velocity data" - 344. "rates" instead "velocities"

We agree that this should be 'calving rate' and changed all occurrences accordingly. We would like to note, however, that often (for close to stationary fronts) front velocities and calving rates are very close and the latter barely distinguishable from the much easier to observe velocity.

- Change the units of calving rate in Table 5 to md-1 in coherence with Figures 13 and 14.

This has been changed in the new version of the manuscript.

**Referee comment Jeremy Bassis (RC2)**

**Major comments**

1. *What is the right stress metric to use and why?* The authors of this study use the Hayhurst criterion, a linear combination of stress tensor invariants. In contrast, Doug Benn and Joe Todd argue that this is a purely empirical relationship and that the largest principle stress is the more physics based metric

and results from a generalization of Nye's zero tensile stress model. Physics tells us that in the absence of anisotropy, the failure criterion should depend on invariants of the stress tensor. However, it is unclear to me how or why physics provides any guidance as to which invariants it depends on. To be more clear, in the uniaxial case, failure is clearly related to the single component of stress and because ice, like most materials, is much weaker in tension than compression observations show that (tensile) failure occurs when the applied strength exceeds some threshold. In the Nye zero stress model, that threshold is set to zero. We know from experiments and field observations that ice has finite strength. Fortunately, Weertman showed that for fields of closely spaced crevasses, the depth of crevasses will approximate the Nye zero stress depth so long as the initial starter crack length is sufficiently large and the strength of ice, measured as critical stress intensity factor, sufficiently small. The generalization to multi-axial failure is less obvious. Benn et al., (2017) apply the maximum principle stress. We did the same thing in Yue and Bassis, (2017), but also considered the possibility that shear failure could occur. Under multi-axial loading there are a larger number of invariants of the stress tensor that must be considered. The Hayhurst criterion attempts to combine multiple modes of failure in a single fracture growth model by taking a linear combination of invariants. Both the Hayhurst and maximum principle stress criterion reduce to the uniaxial case when other stress components vanish and one could view the Hayhurst criterion as a generalization of the Nye criterion to multi-axial loading. The distinction between the two hypotheses is that the maxi- mum principle stress criterion predicts that multi-axial loading will have no effect on the depth or rate of fracture propagation. To interpret the maximum principle stress criterion rigidly, implies that failure of ice only occurs through tensile failure and no other mode of failure is possible, an assertion that is falsified by laboratory measurements. In contrast, the Hayhurst criterion tells us that multi axial loading can increase the depth, trajectory and rate of fracture propagation. Moreover, the Hayhurst criterion reduces to the maximum principle stress criterion and is thus more general. Crucially, I don't see a way to deduce which – if either – is correct in the absence of observations. It is, however, clear that the maximum principle stress criterion has provided useful results that allow us to predict how fields of crevasses respond to changes in stress. That this relation- ship is not deduced from a more fundamental principle doesn't detract from its usefulness. It is also entirely possible that assuming under glaciological regimes tensile failure dominates is a useful approximation. I'm uncomfortable litigating the physical appropriateness of one model versus another without observational data to confirm or refute hypotheses, which is something the authors might think about reviewing. *Where I think this manuscript could improve is to provide a better motivation for why the Hayhurst stress is first introduced and then, why it is abandoned in favor of the maximum principle stress. I think a valuable result would be to show that results are insensitive to the choice of failure metric (Hayhurst or maximum principle stress). We may not know which is the correct law, but it might not matter. Similarly, it would be helpful if the authors could comment on any observational evidence to support or refute the use of the Hayhurst stress in glaciological applications.*

These are good points that are partially discussed in section 3.2, where we note that there is little difference between the Hayhurst stress and the maximum principal stress at the surface. The main difference is that the distance at which the stress maxima are found is located further away for the Hayhurst stress than for the maximum principal stress. As far as we know, there is no observational evidence supporting or refuting the use of the Hayhurst stress, since it is basically an adjusted combination of stress invariants. Pralong and Funk (2005) however determined $\alpha$ and $\beta$ based on laboratory experiments at different temperatures, grain size and specimen size. Further justification for the use of the maximum principal stress for the parametrization has been added in the revised manuscript (lines 250-254).

> *Specifically, we assume that surface crevasses open under the extensional stress $\sigma_1$. The Hayhurst stress would be a similarly suited stress measure for the extensional stress state under small compressive load at the glacier surface. The above stress state analysis showed that the three stress intensity measures $\sigma_1$, $\sigma_e$ and $\chi_H$ along the glacier surface are very similar, as demonstrated in Figure 9.*

2. *Calving rate versus calving positions laws*: As a minor point, the relationship between calving rate and calving position laws is in fact direct (see Bassis, 2010). Calving rate laws can be deduced from

statistical averages over many calving events and are valid over time scales that are much longer than the typical recurrence interval between calving events provided the spatial scale of calving events is small compared to the glacier system. The relationship between the two, in a statistical sense, is really just a switch in time scale. Calving position laws have the advantage that they better encompass fluctuations in calving front position, but are less practical over longer timescales (e.g., millennial) when the only data available is average position at discrete intervals of time. Moreover, many of the 'position laws' can be equivalently formulated as rate laws. A simple example of this is the height-above-buoyancy law and others in its family. It is straightforward to cast this as a continuous rate law for calving front position analogous to those used for grounding line migration. *What is a more fundamental issue for me is whether the terminus position of the glaciers used to calibrate the law are relatively constant or changing. If the glaciers are in or near steady-state than many variables can be correlated without indicating causality. A more convincing argument is if the calving law can predict the rate of retreat/advance for one or more glaciers that is changing.*

Whether a tidewater glacier is in a steady state is difficult to assess. However, the data set we used for calibration contains glaciers that are both in transient (e.g. Columbia 1988) and stationary front positions (e.g. Store). Furthermore, the data set covers the full range of relative water levels and ice thicknesses that are encountered in nature (we now mention this on lines 350-351). As for the choice of a calving rate or a calving position relation, here, we do not argue for or against any relation, as we believe that several types are useful.

3. *Empirical, semi-empirical versus physical calving laws*: Both reviewers brought up the point that the calving law is empirical rather than physical. I have similar concerns, although my physical concerns are slightly different and will be raised in the point that follows. To preface, I do increasingly worry that we are using the concept of 'physical' versus empirical as a blunt cudgel to beat each other. The rheology of ice that we use is an empirical flow law. The formulation presented by Cuffey and Paterson is excellent and results from a set of calibrated model experiments and laboratory measurements. Despite this empirical basis, most of us don't usually describe our ice dynamics models as 'empirical' or 'semi-empirical' due to the fact that the parameters in Glen's flow law are not calculated from first principles. We glaciologists scorn well calibrated empirical data at our peril. Instead of wading into the empirical debate, I would encourage the authors to ask a couple of questions. *(1) What predictions can the calibrated model make that are independent of the data set used to calibrate the model that can be used to falsify the model? The model is calibrated for a suite of Arctic glaciers, but can it be used to predict the calving rates of glaciers that are not included in the data set? For example, there are only a couple of data point for Columbia Glacier, but the retreat has been well documented for several decades. Alternatively, one could use different subsets of data to calibrate the model and then validate against an independent set. More physically, the authors are inferring a threshold stress at which damage begins to grow. This threshold could be compared to observations that indicate the stress at which surface crevasses first appear. (2) How sensitive are the results to the model calibrated parameters? If the model results only weakly depend on the calibrated parameters, then the fact that they are determined empirically is not much of a concern because we only need to get ballpark estimates. However, if the results depend sensitively on one or more parameters than we need to think careful about how to measure these parameters independently and may be concerned that the model predictions may be less reliable when applied elsewhere.*

(1) This is a good point, it would be useful to validate the calving rate predictions by applying this parametrization with an independent set of glaciers. However, the main goal of this part of the study is to be able to estimate calving rates for the selected glaciers using a simple scaling relationship. Furthermore, precise data (calving front height, water level and calving rate) are not available for most glaciers. (2) The results are not very sensitive to the different parameters, thus we think that the use of such a parametrization is not a concern. This is now mentioned in the paper at lines 346-348:

> *Calving rates thus obtained are not very sensitive to the exact choice of parameter values, which are within the range of previous studies (Duddu and Waisman, 2012; Lliboutry, 2002; Vaughan, 1993).*

4. *Physical interpretation of the calving law*: The point that I struggle with the most in this paper is physically interpreting the mathematical model. My interpretation of Equation 22 is that authors are stuffing the maximum principle stress as measured at the surface into their damage evolution law and then evaluating how long it takes for a surface crevasse at the location of the maximum stress to develop. The calving rate is then the distance to the maximum stress at the surface divided by the time scale of the calving event. This calculation, however, seems to give the time scale for a crevasse at the surface to develop and not the time scale for a crevasse to penetrate the entire ice thickness or some fraction thereof. In simulations that we have done using a similar formulation of damage mechanics as presented here, but simulating the propagation of individual crevasses, dry surface crevasses never penetrate the entire ice thickness. I can accept the arguments that lead to a shallow surface crevasse at the surface, but the magic that then asserts that the surface crevasse will penetrate deep enough to cause a calving event is not clear. In fact, looking at the stress field, it looks like the maximum principle (or Hayhurst stress) decreases with depth. Why then does the surface crevasse propagate the entire distance and why doesn't it take longer to propagate down as the stress decreases? It is this step more than the details of the calibration that makes the calving law seem empirical or divorced from 'physics' to me. Getting fractures to propagate the entire ice thickness has always been a problem for calving models and this study seems to sidestep these issues. I guess I'm OK with postulating a calving form base on the position and magnitude of the surface tensile stress, but the departure from physics should be more clearly emphasized. Furthermore, given this departure, I don't quite understand why the complex damage evolution law is used. *What if a linear relationship between damage and principle stress was postulated instead? The rate of damage growth would then have two parameters (a rate factor and stress threshold). Or what if the stress threshold was set to zero, giving a single parameter? Do these choices significantly degrade the fit? Does the calving law depend sensitively on the form of the assumed damage law or is this calibrated out? Can we say anything about the form that the damage evolution law must take if the data is to be matched?*

We clearly stated the assumptions made on crevasse formation in section 4.2, and have now expanded this paragraph to read (lines 268-272):

> *One major assumption is that a large crevasse forms at the location of the maximum tensile surface stress where the ice is weakened until failure. Such crevassing seems realistic as both observations and model results show the formation of huge crevasses (Pralong and Funk, 2005). When failure of the surface ice is complete, all ice in front of the crevasse is removed and a new calving front forms at the location of the crevasse. The processes removing the ice are not specified in this parametrization, but it is assumed that they act on faster time scales than the formation of the surface crevasse.*

The damage relation employed is the classical form of isotropic damage initially proposed by Kachanov (1958). The additional stress threshold had to be included to prevent ice failure under very low stresses.

**Detailed comments**

Page 1 abstract "stress state" or "state of stress"?

This expression was changed in the new version of the manuscript (line 4).

Use of crack: considering using crevasse or defining how the term crack includes more than just crevasses for your glaciological audience.

We think that we used the two terms in their specific meanings in the text.

Page 2, near line 35: "Benn et al. (2007a, b) generalized the flotation criterion by setting the terminus position at the location where crevasses penetrate below the water level" I think the authors are getting at the fact that in the height-above-buoyancy criterion, the position of the calving front is specified rather than a calving rate (as in the Brown et al., water depth model). The Benn et al., approach thus

provides a different model to compute the terminus position based on when surface crevasses penetrate to the water line. This is mechanically different than the height-above-buoyancy criterion, but falls into a similar type of law whereby the position of the calving front is determined.

We agree, this part has been rewritten in the new version of the manuscript (line 35-36):

> *Benn et al. (2007a,b) introduced a physics-based approach by setting the terminus position at the location where crevasses penetrate below the water level.*

Page near line 40: "However, the crevasse depth estimation lacks validation with field observations and is based on a snapshot of the stress balance, neglecting the pre- existence of cracks and their effect on the stress state of the glacier (Krug et al., 2014)." This is an excellent point. Most crevasse penetration models assume that crevasses have a negligible effect on the state of the stress and are purely passive. These crevasse penetration models also ignore the advection of previous existing crevasses into the near terminus region. It is unclear how these effects are incorporated into the proposed model.

These effects are already incorporated in the parametrization (Eq. 21) and affect the time-to-failure. Due to lack of validation data, and the difficulty assigning values of $D_0$ and $D_c$ to real-world glaciers, they are currently lumped into $\tilde{B}$ (Eq. (22)).

Equation (1) is a depth averaged equation. We don't have to rely on it and can instead compute approximations of the state-of-stress using finite element models. However, Equation (1) has the advantage that it is non-parametric (i.e., independent of ice rheology). More sophisticated methods of calculating the stress field require additional, often unknown parameters like the temperature of the ice and an appropriate sliding law.

We agree with this comment.

Page 2 near line 50: "The meaning of such a 'depth averaged' longitudinal stress for local fracture, for example for assessing surface crevasse formation, and the calving processes is not clear." I'm not sure I understand the complaint here. The depth integrated approach yields an estimate of the tensile stress based solely on the ice thickness and water depth. This estimate of stress allows us to estimate the depth of crevasses. Of course, near the terminus the depth integrated formulation may not accurately estimate the stress due to the absence of bending effects and neglected terms in the force balance. This seems like it points to a lack of accuracy rather than a difficulty with interpretation of the meaning of depth averaged stress.

We agree with this comment and have adapted the new version of the manuscript (lines 51-53).

> *However, it should be noted that this vertically integrated stress is not representative for the stress state near the surface of the terminus, and such a 'depth averaged' longitudinal stress may be inaccurate as bending stresses are neglected.*

Page near line 55: "However, observations of the appearance of surface crevasses on glaciers in relation to the strain rate field suggest a much lower cohesive strength of glacier ice between 0.09 and 0.32 MPa (Vaughan, 1993) for cold Antarctic ice streams, or as low as 0.05 MPa for a temperate Alpine glacier (Lliboutry, 2002)." This is an interesting point about the uncertainty in the yield strength, but unless I am misunderstanding, might be related to some confusion about different modes of failure. The fact that surface crevasses are detected at low stresses doesn't imply that the strength of ice in shear must be much lower. Tensile and shear failure can be distinct modes of failure and each can have their own yield strength.

From LEFM, the different fracture modes have different stress intensity and different fracture toughness (not yield strengths). But it is difficult to understand the comment in terms of the discussion in this manuscript. We consider mainly extensional stress/strain rate at the surface which is the same measure determined in the cited papers.

Page 3 near line 70 "This implies that thick glaciers approach flotation at their front but for shallow water depth the constraints on geometry seem less clear." It is true that the bounds are very wide for small ice thicknesses. I would say that bounds are clear, but not particularly useful for small ice thicknesses given the spread of permissible values.

We are not sure we fully understand this argument and tried to reformulate this part in the new version (lines 75-76):

> *This implies that thick glaciers approach flotation at their front but for shallow water depth the bounds on stress, and hence cliff geometry, are less well constrained.*

Equation 3: Should have a dot between the del and u to enforce the divergence of velocity is zero not the gradient of velocity is zero.

We added a dot in the new version of the manuscript (line 97), although contracting the gradient operator with a vector always gives the divergence.

Numerical implementation: How is the incompressibility condition enforced numerically? Is this a mixed element formulation?

Incompressibility was enforced by equation (3). Numerically this was implemented with Q2-Q1 elements, with the relevant text added in lines 105-107:

> *The model domain was discretized with second-order QUAD9 elements with Galerkin weighting. Model variables were $u$, $w$, $\sigma_m$ with a second-order approximation for the velocities and a first-order approximation for the mean stress (forming a LBB-stable set)*

Section 2.2. This might be more clear if one uses the scales introduces to non-dimensionalize the governing equations. Then, I assume, we could write the equations in terms of a set of non-dimensional numbers, like the aspect ration (H/L) that describe the dynamic and geometric similarity between solutions.

We use dimensional units for all calculations as given in the text. We think this is easier to understand, and makes the modeling exercise both more intuitively clear and comparable to other studies.

Section 2.3. I think the boundary condition is traction free not stress free. One is not usually able to prescribe the entire stress tensor.

Yes, a stress tensor projected on a surface is a traction. This was changed in the new version of the manuscript (line 124).

Page 5, line 130: It is also possible to specify zero slip in the vertical direction along the inflow boundary condition. This prevents edge effects near the zero velocity boundary condition.

Many inflow boundary conditions are possible, and all influence the velocity field. The main point here was to obtain an always numerically stable boundary condition (pinning velocity at least at one point).

Results: Section 3.1: It looks like when the authors say "stress" they mean Hayhurst stress, but what parameters were used to calculate the Hayhurst stress? I would have thought that the patterns of, say Von Mises stress would be very different from the largest principle stress? I can't find this information in the text or figure caption. This difficulty in understanding how the Hayhurst stress was calculated continues throughout the rest of the results section. Would it be more more helpful to show the largest principle stress, Von Mises stress in separate panels?

The parameters $\alpha, \beta$ and $\gamma$, which give the relative importance of the 3 stress invariants, used to calculate the Hayhurst stresses in the results section are selected according to Pralong and Funk (2005), as commonly used in previous literature involving the Hayhurst stress applied to ice flow. This

parameter selection was mentioned at line 174 of the old manuscript. For clarity, we introduced this selection of parameters in the introduction of section 3.1 of the new version (lines 173-175)

The maximum principal stress and von Mises stress distributions are now shown in the appendix figure.

Page 7, line 185: The extrusion flow is an example of the Poison Effect.

This is indeed an expression of the *Poisson* effect. We mention it here because of the long history of this topic in glaciology, nicely summarized by Waddington (2010).

Page 8, section 3.1.3. I'm not sure I understand the basal slipperiness results. When we did experiments using a full Stokes model we considered no-slip boundary conditions and free-slip boundary conditions and these two experiments resulted in significant differences in velocity and stress.

According to Weertman theory, a free-slip boundary on an uncoupled ice shelf condition induces a different stress state than a grounded glacier sliding over its bed. Thus, we would obviously expect different behaviours as well in the case of free-slip.

Page 9 line 235: "Figure 9 clearly illustrates that water pressure at the calving front exerts a stabilizing effect on the calving front by both lowering the stresses and decreasing the distance from the calving front at which the stress maximum is located." This is exactly what we argued in Bassis and Walker (2012), although our analysis was less numerically sophisticated.

This is now mentioned in the manuscript (line 244).

Page 9, line 240: Now I think I have lost the thread. Why consider the Hayhurst criterion at all if the maximum principle stress is all that is going to be used? Is the reason the Hayhurst stress is going to be abandoned because all of the invariant combinations give similar answer? This could use a bit more motivation.

In the derivation of the calving relation we use a stress fit along the surface, which happens to have the exceptionally simple form $x \exp(-x)$ for the maximum extensional stress $\sigma_1$ (Eq. (16) and Figure 10). As has been shown in Figure 8 and 9, this stress measure is very similar to the two other measures $\sigma_e$ and $\chi_H$, the latter being a linear combination of the former two. Performing the, somewhat cumbersome, stress-curve fitting exercise on the von Mises stress leads to similar curves, although with slightly different parameters. We clarified this point now in lines 250-254:

> *Specifically, we assume that surface crevasses open under the extensional stress $\sigma_1$. The Hayhurst stress would be a similarly suited stress measure for the extensional stress state under small compressive load at the glacier surface. The above stress state analysis showed that the three stress intensity measures $\sigma_1$, $\sigma_e$ and $\chi_H$ along the glacier surface are very similar, as demonstrated in Figure 9.*

**Short comment by Doug Benn and Joe Todd (SC1)**

**Model formulation**

The Hayhurst stress used in the first part of the analysis is not a physical quantity - it is an ad hoc combination of different stress metrics that - in the absence of any physical understanding of mechanisms - can be tuned to match observations. The Hayhurst stress is, in essence, a semi-empirical approach to divining which factors may control calving behaviour. This notwithstanding, the authors then abandon the Hayhurst stress approach and adopt the maximum principal stress as the foundation of their calving relation. The maximum principal stress is exactly the same stress metric used by Benn et al. (2017), and it is in fact a generalization of the Nye crevasse depth formula used by Benn et al. (2007a). (Martin Lüthi provided a detailed review of Benn et al (2017), so at least one member of the author team has been aware of these results for some time.) The maximum principal stress is

also equivalent, in 2D, to the 'effective stress' metric used by Todd & Christoffersen (2014) to model calving at Store Glacier.

The first aim of this publication heads exactly in the direction pointed out by the comment. We perform an analysis of five possible objective stress measures, and all linear combinations thereof. This is explained in Equation (14), Section 3.2 and Figures 8 and 9. Apparently we did not succeed presenting our findings clearly enough, which we now emphasize in lines 250-254 (see also above):

> *Specifically, we assume that surface crevasses open under the extensional stress $\sigma_1$. The Hayhurst stress would be a similarly suited stress measure for the extensional stress state under small compressive load at the glacier surface. The above stress state analysis showed that the three stress intensity measures $\sigma_1$, $\sigma_e$ and $\chi_H$ along the glacier surface are very similar, as demonstrated in Figure 9.*

Figure 9 shows the envelopes of all objective linear combinations of stress measures that could be used to formulate a fracture or damage evolution relation. One major conclusion is that the two third invariants $I_3$ and $J_3$ play a minor role, and can probably safely be ignored for glacier front geometries, as compared to $\sigma_1$ and $\sigma_e$.

Figure 9 shows that the maximum achievable stress intensity at the surface is the von Mises stress. Deeper in the ice, the Hayhurst stress, as parametrized by Pralong, gives the highest stress intensities.

The authors then formulate a calving rate law using a damage evolution function. Calving rate laws are attractive from a modelling perspective, but their physical justification is unclear. Rate laws are likely valid where calving is driven by melt undercutting, although this process is excluded from the analysis of Mercenier et al. In our work, we have focused on calving position laws, which predict the location of the calving front from the state of stress at any given time. Model experiments with the discrete element model HiDEM provide justification for this approach, because calving events occur rapidly in response to specific states of stress (Benn et al., 2017). Given the existence of these contrasting approaches to formulating calving laws, we feel that some discussion of this issue would benefit the present paper, and ideally the authors should provide more detailed justification for choosing a rate law.

As mentioned before, in this work, we do not argue for or against a type of calving relation that should be used. We believe that several calving relations types can be useful for different purposes. However, explanations for the use of a calving rate relation have been added at lines 339-343:

> *Thus, calving rates depend on the surface evolution and hence the upstream dynamics of the glacier. The semi-empirical calving rate parameterization is therefore, in the sense of inclusion of upstream dynamics, similar to the position based calving models (Benn et al., 2007a; Nick et al., 2010; Todd and Christoffersen, 2014; Benn et al., 2017). The formulation as a calving rate also makes this parametrization relatively easy to use in larger-scale fixed grid models.*

In lines 317-8, the authors claim that their proposed calving parametrization is "physics based, as opposed to the purely or semi-empirical nature of other approaches". This is wrong on two counts. First, crevasse depth calving laws (Benn et al., 2007; Nick et al., 2010; Todd and Christoffersen 2014; Benn et al. 2017) are physics based. Second, the calving parameterization proposed by Mercenier et al. is itself 'semi-empirical', and is reliant on tuning to data.

We agree with the above argumentation and have adapted the new version of the manuscript (lines 339-343, cited just above, and not repeated here).

**Model predictions and observations**

In Section 5.3, the authors obtain values for two empirical parameters $B$ (damage evolution rate) and $\sigma_{th}$ (damage threshold) using data from calving glaciers in the Arctic. The text in this section is rather obscure, but it seems that the data plotted in Figs. 13 and 14 are the same as those used for

model tuning. Thus, the calving data in Fig 13 (wrongly described as 'velocity data' on line 340) are not shown 'for comparison', but are in fact the data points used to tune the position of the isolines of calving rate. Furthermore, Figure 14 does not compare calving data with model predictions, but compares calving data with predictions from a calving law tuned using the same data. It is therefore a representation of model fit rather than model performance. The authors are not justified in claiming that the model "predicts calving rates. . . reasonably well" (line 355), or that it is "in good agreement with observations" (line 10). The results simply mean that it is possible to tune the model to fit the data, not that the model has actual predictive power.

Calving rates have very rarely been measured directly, but are inferred, as in our case, by the rate of ice motion (velocity) and the rate of front retreat, which is much smaller than terminus change.

As for the rest of the comment, we see the point the authors make, and partially agree. Section 5.3, conclusions and abstract statements about the model calibration and performance have been adapted in the new version of the manuscript.

**Errors in data plotting**

Equation 22 and Figures 11, 13 and 14 do not seem to show the same calving parameterization. For example, plugging the "Columbia 2000" data into Eq. 22 (H = 382m, Hw = 260m, w = 0.68, measured Uc = 24.7 m d-1), with B = 37 MPa-r a-1, $\sigma_{th}$ = 0.17 MPa, r = 0.43, $\sigma_{ref}$ = 0.009H, gives a calving rate of 20.72 m d-1. However, in Figure 11, taking w $\sim$ 0.7, and ice thickness = 382m, calving rate is > 23 m d-1. In Figure 13, the intersection between H = 382m and w = 0.67 gives a value greater than 45 m d-1. Finally, in Figure 14, 'Col 2000' is shown with a predicted calving rate of $\sim$ 36 m d-1.

This is a very good catch, thank you. Actually, some of the figures still showed results obtained with an older set of parameters: Figures 11, 12 and 13 were produced with a different value of $\tilde{B}$ than the final value. This has now been corrected in the new version of the manuscript.

**Review of previous literature**

Previous literature is cited in a very partial way, and some previous work is misrepresented. For example (line 34) the authors state that Benn et al. (2007a, b) "generalized the flotation criterion", and later (line 318) they imply that these papers take an empirical or semi-empirical approach to modelling calving. These statements are untrue. The papers by Benn et al. proposed a new, physically based approach to modelling calving, setting the position of the calving front where crevasses penetrate to the waterline (Benn et al., 2007a). This was later modified by Nick et al. (2010) to include crevasse penetration through the full thickness of the glacier. These initial formulations computed crevasse depths from only longitudinal stresses, but subsequent work has generalised the crevasse criterion to include extensional stress in 2D (Todd and Christoffersen, 2014) and 3D (Benn et al., 2017). Benn et al. (2017) also discussed at length the issue of stress balance "snapshots", and proposed strategies for overcoming these limitations. The criticism that crevasse depth models lack "validation with field observations" (line 39) is also unwarranted. Some authors have tuned the model to match observations (e.g. Nick et al., 2014; Lea et al., 2014), which is exactly the same approach as taken by Mercenier et al. Comparison of the predictions of an untuned crevasse depth model against independent observations has been done by Todd et al. (in press), although of course Mercenier et al. cannot be expected to cite this work. A copy can be supplied on request.

We agree with this comment and adjusted the review of the literature accordingly in the new version of the manuscript (lines 35-45):

*Benn et al. (2007a,b) introduced a physics-based approach by setting the terminus position at the location where crevasses penetrate below the water level. The crevasse depth is computed using the Nye (1957) theory which relies on the equilibrium between longitudinal stretching and overburden stress of the ice. This dynamic approach for calving allowed for successful reproduction of calving front variations of ocean-terminating glaciers in Greenland and Antarctica (Nick et al., 2010; Otero et al., 2010; Nick et al., 2013; Cook et al., 2014; Otero et al., 2017). Although the crevasse depth model can be calibrated to observations (Lea et al., 2014), it lacks validation with field observations and is based on a snapshot of the stress balance, neglecting the pre-existence of cracks and their effect on the stress state of the glacier (Krug et al., 2014). A recent, more sophisticated approach by Benn et al. (2017) predicts calving positions based on the maximum principal stress distribution, and accounts for the effect of water pressure in the submerged parts of the glacier front by combination of a continuum flow model with a discrete element model to simulate calving events.*

In lines 42-50, the authors present an approximation for depth averaged longitudinal stress and state that it is the "main driving force" of crevasse depth models. However, the crevasse depth criterion is not fundamentally a depth-averaged law, although it has previously been implemented in 1D dynamic models (e.g. Nick et al. 2010). Todd and Christoffersen (2014) implemented the crevasse depth criterion in a 2D full-Stokes model, similar to that presented by Mercenier et al., and computed crevasse penetration locally based on nodal stresses. The version of the crevasse-depth calving law adopted by Benn et al. (2017) uses the maximum principal stress, which is exactly the same metric used by Mercenier et al. in the second part of their analysis.

This statement applies generally to steep calving fronts and is not related to any model. We make this distinction clearer by reformulating it. (lines 46-47):

*For near-vertical calving fronts, the main driver for calving is the horizontal deviatoric stress $\sigma'_{xx}$ in vicinity of the laterally-confined calving front.*

The review of melt undercutting (lines 63-67) also misrepresents the literature. Hanson and Hooke (2000) did not look at undercutting at all, and O'Leary and Christoffersen did not "suggest that an increase of water depth leads to a higher rate of oversteepening development". In fact, O'Leary and Christoffersen argued that increasing water depth increased the stress response to a given amount of undercutting, and hence the magnitude of the 'calving multiplier' effect. The opposite trend was found in the more detailed model experiments by Benn et al. (2017), who showed that the effect of undercutting is diminished by increasing water depth. Cook et al (2014) and Krug et al (2015) did indeed conclude that melt undercutting does not significantly affect calving rates. However, this is due to the insensitivity of their models to undercutting, as discussed in detail by Benn et al. (2017). In fact, recent studies have shown that melt undercutting is the primary driver of calving in Svalbard and some Greenland fjords (Luckman et al., 2015; Cowton et al., 2016).

The aim of this paragraph was to look at the over-steepening effect rather than the melt undercutting process. However, while they do not mention the melt undercutting process directly, the Hanson & Hooke reference refers to the over-steepening of the calving face, which can be caused by melt undercutting or a high water depth. The context being that it is over-steepening that causes the destabilization of the calving face, we consider this reference to be appropriate. We tried to clarify this section in the new version of the manuscript (lines 65-71).

*Calving termini can also be over-steepened by melt undercutting, which leads to higher stress intensities (Hanson and Hooke, 2000) and may facilitate calving (Benn et al., 2017). Ice flow model results (Hanson and Hooke, 2000) suggest that an increase of water depth leads to a higher rate of over-steepening development at the calving front and thus an increase of calving activity. However, model results seem to indicate that melt undercutting does not significantly affect calving rates (Cook et al., 2014; Krug et al., 2015) while other studies suggest that calving rates are strongly related to melt undercutting for some arctic glaciers (Luckman et al., 2015; Petlicki et al., 2015; Cowton et al., 2016). Conversely, a calving front inclined towards the inland is expected to be more stable than a vertical cliff.*

**Additional point**

Regarding the role of basal sliding, the authors provide a useful perspective on their results in lines 303-4, in which they point out that spatial variations in basal slipperiness (as would result from a pressure-dependent sliding law) would likely introduce velocity gradients that could affect calving. This caveat is not reflected in the statements in the abstract ("the effect from VARIATIONS in basal sliding is much smaller" (emphasis added) and the conclusions ("basal sliding likely has a weaker effect ...on stability": lines 353-4). Because they only impose uniform basal slipperiness, the experiments presented in this paper cannot evaluate the influence of basal sliding on calving - and this includes their important relationship with water depth. The summary statements in the abstract and conclusions should reflect this.

This is a good point, the new version of the manuscript is adapted by mentioning that the variations in slipperiness are spatially uniform (lines 6 and 369-370).

> *Results show that water depth and calving front slope strongly affect the stress state while the effect from spatially uniform variations in basal sliding is much smaller.*

> *Spatially uniform variations in basal sliding likely have a weaker effect than water depth and calving front slope on the stability.*

**Concluding remarks**

The proposed calving relation has the benefit of simplicity, and makes some interesting and testable predictions. However, we are sceptical this it represents an improvement on existing approaches because it is based on only one control on calving - the stresses introduced by the force imbalance at the ice front - and neglects other important processes. Calving also occurs in response to longitudinal stresses caused by along-flow variations in basal and lateral drag; melt undercutting in response to heat flux from the ocean; and super-buoyancy where glaciers flow rapidly into deep water. Indeed, these processes are known to be the main drivers of calving on several of the glaciers listed in Table 5. The predictions of the model listed in lines 321-326 correspond to some observed behaviour (e.g. the increasing instability of high, unsupported ice cliffs) but not others (e.g. calving triggered by ice flow into deepening water).

We fully agree with these remarks. At the same time we doubt that any simple calving relation exists that includes all of these effects, all of which are difficult to observe and to quantify.

**References**

[revised manuscript text omitted]

---

## Author Response (AR2)

We would like to thank the referee Jeremy Bassis for his renewed work on this manuscript. Below, all points raised (in black) are addressed (with responses in blue) and changes to the text are *emphasized* after each point. Changes to the manuscript are written in **red** in the new version.

**1 Major Point of Confusion:**

I'm still confused by the discussion related to the analytic calving relation in section 4.2 My understanding is that the authors are stuffing their parameterized estimate for the principle stress at the surface of the glacier into Equation (20) and integrating to find how long it takes damage to accumulate from an initial value to a critical value. So far so good. The 'calving rate' is then estimated as the distance to the stress maximum divided by the time for damage to accumulate to the critical value. This is where I'm confused and I have two problems with this calculation (and I apologize if these stem from my own misunderstanding). The first problem is that the time-to-failure calculated is the time-to-failure of a piece of ice (or element in the FEM formulation) at the surface of the glacier. This calculation, as the authors note, tells us how long it takes for a crevasse to form near the surface of the glacier. However, this calculation **doesn't** tell us how long it takes for a crevasse to penetrate the entire ice thickness (or some fraction thereof). To calculate that one would presumably require an average vertical crevasse penetration velocity. The average vertical crevasse penetration velocity times the ice thickness (or fraction thereof) would then give the time-to-failure. Crudely speaking, we might roughly say that the vertical crevasse penetration velocity is analogous to Equation 19, but with a length scale given by the size of the nodes in the FEM model. This would result in a calving rate that includes an additional factor of ice thickness: it takes longer for a crevasse to penetrate the entire thickness if the glacier is thicker. It is possible that the authors are assuming that crevasse penetrate the entire ice thickness (or fraction thereof) as soon as the crevasse forms at the surface, but this was not clearly stated anywhere I could find. Moreover, because the tensile stress clearly decreases with depth, this assumption is tenuous and full-thickness calving will not occur just because a surface crevasse appears. (There are many observations of surface crevasses not associated with calving events). Furthermore, FEM models with damage computed similarly to that assumed here show that surface crevasses only penetrate a fraction of the depth. Fundamentally, the requirement that a surface crevasse forms seems to be an insufficient requirement for calving and I don't understand how this gives you a calving rate. In general, I'm not opposed to heuristically arguing that one might attempt to fit an empirical law to the data available, but I think readers need to know what the assumptions are and what is "physics" and what is extrapolation/guess work.

Perhaps our description of the assumptions made for the calving relation was not clear enough. The assumptions were stated in the revised paper as (lines 266ff):

*One major assumption is that a large crevasse forms at the location of the maximum tensile surface stress where the ice is weakened until failure. Such crevassing seems realistic as both observations and model results show the formation of huge crevasses (Pralong and Funk, 2005). When failure of the surface ice is complete, all ice in front of the crevasse is removed and a new calving front forms at the location of the crevasse. The processes removing the ice are not specified in this parametrization, but it is assumed that they act on faster time scales than the formation of the surface crevasse.*

We do agree that the tensile stress decreases with depth and thus cannot be responsible for the penetration of a crevasse through the whole thickness. In order to clarify this issue, we extended this paragraph by mentioning possible processes that might drive the downwards propagation of a crevasse. We assume that the relevant timescale for these processes is either much shorter, or proportional to the time-of-failure calculated with the damage relation. This paragraph now reads as follows (lines 261-270):

*One major assumption is that a large crevasse forms at the location of the maximum tensile surface*

*stress where the ice is weakened until failure. Such crevassing seems realistic as both observations and model results show the formation of huge crevasses. When failure of the surface ice is complete, we assume that all ice in front of the crevasse is removed and a new calving front forms at the location of the crevasse. Here, we do not consider explicitly which processes are responsible for the rapid removal of the ice downstream of the crevasse. Several processes could be considered, such as bottom crevassing, hydro-fracturing by ponding water in surface crevasses, rapid elastic crevasse propagation (Krug et al., 2014), ice break-off in multiple steps (e.g. a surface slump, followed by subaqueous buoyant calving), or continued material fatigue due to tidal forcing. The proposed calving relation relies on the major assumption that processes responsible for ice break-off act on faster time scales than the formation of the surface crevasse, and therefore that the calving process is uniquely determined by the time to failure at the surface stress maximum.*

**2  Minor points:**

Page 2, near line 55: I still think this is confusing two separate things: the yield strength of ice depends on the mode of failure. The authors are confusing tensile failure with shear failure. Observations indicate that the strength of ice is different for these two modes.

We agree with this comment and have removed the references to ice strength.

Equation 2 is wrong. The momentum equation is related to the **divergence** of the Cauchy stress tensor not the gradient. This also relates to a response to one of my previous comments where the authors erroneously state that the gradient of a vector is the divergence. The gradient of a vector is tensor. In fact the symmetric part of the gradient of the velocity is the rate of deformation tensor, which for small strains is equivalent to the strain rate tensor. The divergence of the velocity is a scalar and is related to incompressibility. Similarly, the divergence of the stress tensor is a vector. The gradient of the stress tensor, however, is a higher-order tensor. Near Equation 4: You can always decompose the Cauchy stress into an isotropic and deviatoric component. This has nothing to do with incompressibility. For example, elastic materials are not incompressible and the stresses can be decomposed into deviatoric and isotropic.

We agree that there was a typo (missing dot) and now write the divergence operator as "div".

Page 4, near line 115: I think "cartesian" should be capitalized??

This has been corrected.

Equation 11 is, I believe a condition on the traction at the calving front. I think you require a condition on the dot product of the stress tensor with the outward pointing normal vector. The normal component is, as the authors state, equivalent to the normal pressure of water, but you also need to enforce that the tangential traction vanishes.

Yes, the tangential stress is zero. Therefore there is nothing to enforce in the implementation in the FE-code. To avoid ambiguities, Equation (11) has been adapted in the new version of the manuscript.

**References**

[revised manuscript text omitted]

---

## Author Response (AR3)

Dear Olivier,

Thank you for taking the time to edit this manuscript. Your comments and guidance have helped improve this manuscript considerably. Below, all points raised (in black) are addressed (with responses in blue). All the suggested changes have been included in the new version of the manuscript.

Some typos / corrections:
- page 2, line 47: would be nice to specify page number when referencing to a book (here at some other places; e.g. line 115)

The page number references have been added.

- Eq. (1): would the sign $\simeq$ instead of = be more appropriate here? To my understanding it is an approximation of the stress, not its exact value.

We agree, the = sign has been changed to $\simeq$.

- page 4, line 10: QUAD9 is a bit specific to the library you are using. Quadrangle with 9 nodes would be more general?

We agree, we replaced "QUAD9" by "9 node quadrangle" at line 100.

- line 143: see Fig. 2 for illustration. (and at many other places in the text, check this and refer to TC rules).

Figure and equation references have been updated according to the TC rules.

- line 156: $J_2$ and $\sigma_e$ are used alternatively in the text and figure caption, which is a bit confusing. Can you chose one notation for the von Mises stress and keep to it? For example, line 229, in the text it is $J_2$ (without hat) and $\sigma_e$ in the legend of Fig. 8 (with an hat). Should be the same.

$\sigma_e$ is now the notation used in the whole manuscript. However, in the legend of Fig. 8, the hat is relevant as it is a scaled version of the von Mises stress.

- line 265: "removal of the ice downstream". I think what you mean is the downstream propagation of a crevasse? There is no ice removal here, just the propagation of the opening of the crevasse?

We agree, this was not clear, "removal of the ice downstream" has been replaced by "downward propagation of the crevasse" (lines 264-265).